# SE(3) Equivariant Augmented Coupling Flows

**Laurence I. Midgley**[*]
University of Cambridge
lim24@cam.ac.uk

**Vincent Stimper**[*]
Max Planck Institute for Intelligent Systems
University of Cambridge
vs488@cam.ac.uk

**Javier Antorán**[*]
University of Cambridge
ja666@cam.ac.uk

**Emile Mathieu**[*]
University of Cambridge
ebm32@cam.ac.uk

**Bernhard Schölkopf**
Max Planck Institute
for Intelligent Systems
bs@tue.mpg.de

**José Miguel Hernández-Lobato**
University of Cambridge
jmh233@cam.ac.uk

## Abstract

Coupling normalizing flows allow for fast sampling and density evaluation, making them the tool of choice for probabilistic modeling of physical systems. However, the standard coupling architecture precludes endowing flows that operate on the Cartesian coordinates of atoms with the SE(3) and permutation invariances of physical systems. This work proposes a coupling flow that preserves SE(3) and permutation equivariance by performing coordinate splits along additional augmented dimensions. At each layer, the flow maps atoms' positions into learned SE(3) invariant bases, where we apply standard flow transformations, such as monotonic rational-quadratic splines, before returning to the original basis. Crucially, our flow preserves fast sampling and density evaluation, and may be used to produce unbiased estimates of expectations with respect to the target distribution via importance sampling. When trained on the DW4, LJ13, and QM9-positional datasets, our flow is competitive with equivariant continuous normalizing flows and diffusion models, while allowing sampling more than an order of magnitude faster. Moreover, to the best of our knowledge, we are the first to learn the full Boltzmann distribution of alanine dipeptide by only modeling the Cartesian positions of its atoms. Lastly, we demonstrate that our flow can be trained to approximately sample from the Boltzmann distribution of the DW4 and LJ13 particle systems using only their energy functions.

## 1 Introduction

Modeling the distribution of a molecule's configurations at equilibrium, known as the Boltzmann distribution, is a promising application of deep generative models [Noé et al., 2019]. While the unnormalized density of the Boltzmann distribution can be obtained via physical modeling, sampling from it typically requires molecular dynamics (MD) simulations, which are expensive and produce correlated samples. A promising alternative is to rely on surrogate deep generative models, known as Boltzmann generators. We can draw independent samples from these and debias any expectations

---

[*]Equal contribution
Code: https://github.com/lollcat/se3-augmented-coupling-flows

37th Conference on Neural Information Processing Systems (NeurIPS 2023).

estimated with the samples via importance weighting. The Boltzmann distribution typically admits rotation and translation symmetries, known as the Special Euclidean group SE(3), as well as permutation symmetry. These constraints are important to incorporate into the model as they improve training efficiency and generalization [Cohen and Welling, 2017, Batzner et al., 2022, Köhler et al., 2020]. Other key desiderata of Boltzmann generators are that they allow for fast sampling and density evaluation. These are necessary for energy-based training, that is, training using the Boltzmann distribution's unnormalized density [Noé et al., 2019, Stimper et al., 2022, Midgley et al., 2023]. Training by energy is critical, as it prevents the model quality from being constrained by the quality and quantity of MD samples.

Existing coupling flows which approximate the Boltzmann distribution of molecules are at least partially defined over internal coordinates, i.e. bond distances, bond angles, and dihedral angles [Wu et al., 2020, Campbell et al., 2021, Köhler et al., 2023a, Midgley et al., 2023], which are SE(3) invariant. However, the definition of these depends on the molecular graph and they are non-unique for most graphs. Furthermore, models parameterized in internal coordinates struggle to capture interactions among nodes far apart in the graph and cannot capture the permutation invariance of some atoms. Thus, they are not suitable for particle systems such as LJ13 [Köhler et al., 2020]. SE(3) equivariance constraints have been applied to continuous normalizing flows (CNFs) operating on Cartesian coordinates [Köhler et al., 2020, Satorras et al., 2021a], and to the closely related diffusion models [Xu et al., 2022, Hoogeboom et al., 2022, Yim et al., 2023]. These models are built upon SE(3) equivariant graph neural networks (GNNs) [Satorras et al., 2021b, Geiger and Smidt, 2022, Batzner et al., 2022]. These architectures can be applied to any molecular graph [Jing et al., 2022], enabling a single generative model to generalize across many molecules. Alas, sampling and evaluating the density of CNFs and diffusion models typically requires thousands of neural network evaluations [Xiao et al., 2022], preventing them from being trained by energy. As such, presently no Boltzmann generator exists that (i) acts on Euclidean coordinates of atoms (ii) enforces SE(3) equivariance, and (iii) allows for fast sampling.

To address this gap, we propose a flexible SE(3) equivariant coupling flow that operates on the Cartesian coordinates of atoms, allowing for fast sampling and density evaluation. Our contributions are:

- We extend coupling layers to be SE(3) equivariant by augmenting their input space with auxiliary variables [Huang et al., 2020] which can be acted upon on by SE(3). We update the atom positions conditioned on the auxiliary variables by first projecting the atoms into an SE(3)-invariant space and then applying a standard normalizing flow transform before projecting its output back onto the equivariant space.

- We demonstrate that, when trained by maximum likelihood, our flow matches the performance of existing SE(3) CNFs and diffusion models as well as coupling flows operating on internal coordinates on molecular generation tasks. Our flow is more than 10 times faster to sample from than SE(3) CNFs and diffusion models. Concurrently with Klein et al. [2023b], we are the first to learn the full Boltzmann distribution of alanine dipeptide solely in Cartesian coordinates.

- We demonstrate our flow in the energy-based training setting on DW4 and LJ13, where parameters are learned using only the molecular energy function. Energy-based training of CNFs or diffusion models is intractable due to slow sampling and density evaluation. Flows operating on internal coordinates are not able to capture the permutation invariance of these problems. Hence, our flow is the only existing permutation and SE(3) equivariant method that can tractably be applied there.

## 2 Background: coupling flows and invariant models

### 2.1 Normalizing flows and coupling transforms

A (discrete-time) normalizing flow is a flexible parametric family of densities on $\mathcal{X}$ as the push-forward of a base density $q_0$ along an invertible automorphism $f_\theta : \mathcal{X} \to \mathcal{X}$ with parameters $\theta \in \Theta$ [Papamakarios et al., 2021]. The density is given by the change of variable formula:

$$q_\theta(x) = q_0(f^{-1}(x)) \left| \det \frac{\partial f_\theta^{-1}(x)}{\partial x} \right|. \tag{1}$$

We can efficiently sample from the flow by sampling from $q_0$ and mapping these samples through $f_\theta$ in a single forward pass. A popular way to construct $f_\theta$ is to use coupling transforms. The $D$

dimensional input $x \in \mathcal{X}$ is split into two sets, transforming the first set conditional on the second, while leaving the second set unchanged:

$$
\begin{aligned}
y_{1:d} &= \mathcal{T}(x_{1:d}; x_{d+1:D}), \\
y_{d+1:D} &= x_{d+1:D}.
\end{aligned}
\tag{2}
$$

They induce a lower triangular Jacobian, such that its determinant becomes $|\partial \mathcal{T}(x_{1:d}; x_{d+1:D})/\partial x_{1:d}|$. Further, choosing $\mathcal{T}$ to have an easy to compute determinant, such as an elementwise transformation [Dinh et al., 2015, 2017, Durkan et al., 2019], allows for fast density evaluation and sampling at low computational cost.

## 2.2 Equivariance and invariance for coupling flow models of molecular conformations

Throughout, we deal with observations of an $n$-body system represented by a matrix $x = [x^1, \ldots, x^n] \in \mathcal{X} = \mathbb{R}^{3 \times n}$, where the rows index Cartesian coordinates and the columns index individual particles. We seek to construct flows on $\mathcal{X}$ endowed with the symmetries present in molecular energy functions. These are invariant to rotations and translations of $x$ (SE(3)), and to permutation of atoms of the same type ($S_n$). We will formalize them in this section.

**Symmetry groups**  The special Euclidean group SE(3) is the set of orientation preserving rigid transformations in Euclidean space. Its elements $t \in$ SE(3) can be decomposed into two components $t = (R, u)$ where $R \in$ SO(3) is a $3 \times 3$ rotation matrix and $u \in \mathbb{R}^3$ represents a translation; for a coordinate $v \in \mathbb{R}^3$, $t \cdot v = Rv + u$ denotes the action of $t$ on $v$. The symmetric group $S_n$ defined over a set of $n$ atoms consists of all $n!$ permutations that can be performed with said atoms. Its elements $\sigma \in S_n$ act on an $n$-body system as $\sigma \cdot x = [x^{\sigma(1)}, \ldots, x^{\sigma(n)}]$.

**Equivariant maps**  A map $f : \mathcal{X} \to \mathcal{Y}$ is said to be *G-equivariant* if it commutes with the group action $\cdot$, i.e. if for any $x \in \mathcal{X}$ and $g \in G$ we have $f(g \cdot x) = g \cdot f(x)$. Invariance is a special case where for any $x \in \mathcal{X}$ and $g \in G$ we have $f(g \cdot x) = f(x)$. There has been a plethora of recent work on constructing graph neural network functions equivariant to the action of $G = $ SE(3) $\times S_n$ [e.g. Thomas et al., 2018, Satorras et al., 2021b, Geiger and Smidt, 2022], which we will leverage to construct our equivariant coupling flow model.

**Invariant density**  A density $p : \mathcal{X} \to \mathbb{R}_+$ is $G$-invariant if for any $x \in \mathcal{X}$ and $g \in G$ we have $p(g \cdot x) = p(x)$. Combining an invariant base density $q_0$ with an equivariant invertible transform $f$, as in (1), yields an invariant flow density [Papamakarios et al., 2021, Köhler et al., 2020]. This gives a practical way to design invariant densities models.

**Challenges in constructing SO(3) $\times S_n$ invariant flow models**  Unfortunately, no coupling transform can be simultaneously equivariant to both permutation of the particles and their rotations; coupling splits must be performed either across particles or spatial dimensions which would break either permutation or rotational symmetry [Köhler et al., 2020, Bose et al., 2022].

Furthermore, there does not exist a translation invariant *probability* measure, as any such measure would be proportional to the Lebesgue measure and therefore not have unit volume. This precludes us from defining an invariant base distribution directly on $\mathcal{X}$. Fortunately, Proposition A.1 and its converse allow us to disintegrate the probability measure into a translational measure proportional to the Lebesgue measure and an SO(3)-invariant probability measure on the subspace of $\mathbb{R}^{3 \times n}$ with zero center of mass. We can drop the former and only model the latter.

## 3  Method: SE(3) $\times S_n$ equivariant augmented coupling flow model

This section describes our main contribution, an SE(3) $\times S_n$ equivariant coupling flow. We first lay the groundwork for achieving translation invariance by defining our flow density on a lower-dimensional "zero Center of Mass (CoM)" space. To preserve permutation and rotation equivariance, we leverage the augmented flow framework of Huang et al. [2020]. Specifically, we use sets of augmented variables as a pivot for coupling transforms. Sec. 3.1 introduces a novel class of coupling transforms that achieve the aforementioned permutation and rotation equivariance by operating on atoms projected using a set of equivariant bases. Sec. 3.2 describes our choice of invariant base distribution and, finally, in Sec. 3.3, we discuss several schemes to train the augmented flow from either samples or energy functions and how to perform efficient density evaluation.

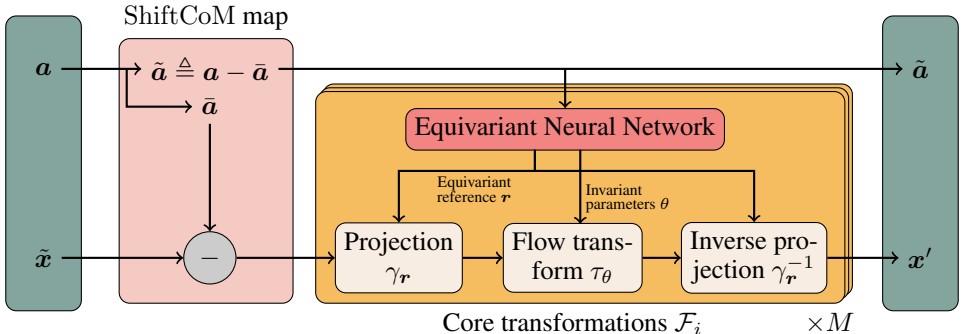

Figure 1: Illustration of the equivariant coupling layer of our augmented normalizing flow, where our variable with zero center of mass (CoM) $\tilde{x}$ is transformed with the augmented variable $a$.

**Translation invariance** is obtained by modelling the data on the quotient space $\mathbb{R}^{3 \times n} / \mathbb{R}^3 \triangleq \tilde{\mathcal{X}} \subseteq \mathcal{X}$, where all $n$-body systems that only differ by a translation are "glued" together, i.e. where $x \sim x'$ if $x = x' + p$ with $p \in \mathbb{R}^3$. Constructing a parametric probabilistic model over $\tilde{\mathcal{X}}$, automatically endowing it with translation invariance. In practice, we still work with Cartesian coordinates, but center the data so as to zero its CoM: $\tilde{x} \triangleq x - \bar{x}$ with $\bar{x} \triangleq \frac{1}{n} \sum_{i=1}^{n} [x]^i$. Thus, $\tilde{x}$ lies on $\tilde{\mathcal{X}}$, an $(n-1) \times 3$ dimensional hyperplane embedded in $\mathcal{X}$.

**Augmented variable pivoted coupling** To allow for simultaneously permutation and rotation equivariant coupling transforms, we introduce *augmented* variables $a \in \mathcal{A}$. Our coupling layers update the particle positions $x$ conditioned on $a$ and vice-versa. The augmented variables need to "behave" similarly to $x$, in that they can also be acted upon by elements of $\mathrm{SE}(3) \times S_n$. We achieve this by choosing $a$ to be a set of $k$ of observation-sized arrays $\mathcal{A} = \mathcal{X}^k$, which we will discuss further in App. B.4. Importantly, we do not restrict $a$ to be zero-CoM.

**Invariant flow density on the extended space** We parameterize a density $q$ over the extended space $\tilde{\mathcal{X}} \times \mathcal{A}$ w.r.t. the product measure $\lambda_{\tilde{\mathcal{X}}} \otimes \lambda_{\mathcal{A}}$, where $\lambda_{\tilde{\mathcal{X}}} \in \mathcal{P}(\tilde{\mathcal{X}})$ and $\lambda_{\mathcal{A}} \in \mathcal{P}(\mathcal{A})$ respectively denote the Lebesgue measure on $\tilde{\mathcal{X}}$ and $\mathcal{A}$. We use $q(x, a)$ as a shorthand for the density of the corresponding zero-CoM projection $q(\tilde{x}, a)$. The density $q$ is constructed to be invariant to the action of $G \triangleq \mathrm{SO}(3) \times S_n$ when simultaneously applied to the observed and augmented variables, that is, $q(x, a) = q(g \cdot x, g \cdot a)$ for any $g \in G$. We aim to construct a $G$-equivariant flow $f : \tilde{\mathcal{X}} \times \mathcal{A} \to \tilde{\mathcal{X}} \times \mathcal{A}$ on this extended space, and combine it with $q_0 : \tilde{\mathcal{X}} \times \mathcal{A} \to \mathbb{R}^+$, a $G$-invariant base density function, to yield the invariant flow density

$$q(x, a) = q_0(f^{-1}(x, a)) \left| \det \frac{\partial f^{-1}(x, a)}{\partial(x, a)} \right|. \tag{3}$$

**Proposition 3.1** (Invariant marginal). *Assume $q : \mathcal{X} \times \mathcal{A} \to \mathbb{R}_+$ is a $G$-invariant density over the probability space $(\mathcal{X} \times \mathcal{A}, \lambda_{\mathcal{X}} \otimes \lambda_{\mathcal{A}})$, then $q_x \triangleq \int_{\mathcal{A}} q(\cdot, a) \lambda_{\mathcal{A}}(\mathrm{d}a) : \mathcal{X} \to \mathbb{R}_+$ is a $G$-invariant density w.r.t. to the measure $\lambda_{\mathcal{X}}$.*

*Proof.* For any $g \in G$, $x \in \mathcal{X}$ and $a \in \mathcal{A}$

$$q_x(g \cdot x) = \int_{\mathcal{A}} q(g \cdot x, a) \lambda_{\mathcal{A}}(\mathrm{d}a) = \int_{g^{-1} \mathcal{A}} q(g \cdot x, g \cdot a) \lambda_{\mathcal{A}}(\mathrm{d}\, g \cdot a) = \int_{\mathcal{A}} q(x, a) \lambda_{\mathcal{A}}(\mathrm{d}a) = q_x(x),$$

where we used the $G$-invariance of $q$ and of the measure $\lambda_{\mathcal{A}}$, as well as $g^{-1} \mathcal{A} = \mathcal{A}$. $\square$

## 3.1 SE(3) and permutation equivariant coupling transform

We now derive our $\mathrm{SE}(3) \times S_n$ equivariant map $f : \tilde{\mathcal{X}} \times \mathcal{A} \to \tilde{\mathcal{X}} \times \mathcal{A}$ defined on the extended space. We introduce two modules, a shift-CoM transform, swapping the center of mass between the observed and augmented variables, and an equivariant core transformation, which updates $\tilde{x}$ conditional on $a$ and vice versa. Composing them yields our equivariant coupling layer illustrated in Fig. 1.

**SE(3) × $S_n$ equivariant coupling** We dissociate the equivariance constraint from the flexible parameterized flow transformation by (1) projecting the atoms' Cartesian coordinates into a learned,

local (per-atom) invariant space, (2) applying the flexible flow to the invariant representation of the atoms' positions, and (3) then projecting back into the original Cartesian space. Specifically, we construct a core coupling transformation that composes (a) an invariant map $\gamma : \Xi \times \mathcal{X} \to \mathcal{Y}$, where $\mathcal{Y}$ is isomorphic to $\mathcal{X}$, and $r \in \Xi$ parametrizes the map. We denote the parametrized map as $\gamma_r$. It is followed by (b) a standard flexible normalizing flow transform, e.g. a neural spline, $\tau : \mathcal{Y} \to \mathcal{Y}$, and (c) the inverse map $\gamma^{-1}$. Denoting the inputs with superscript $\ell$ and with $\ell + 1$ for outputs, our core transformation $\mathcal{F} : (\boldsymbol{x}^\ell; \boldsymbol{a}^\ell) \mapsto (\boldsymbol{x}^{\ell+1}, \boldsymbol{a}^{\ell+1})$ is given as

$$
\begin{aligned}
\boldsymbol{x}^{\ell+1} &= \gamma_r^{-1} \cdot \tau_\theta(\gamma_r \cdot \boldsymbol{x}^\ell), \quad \text{with} \quad (\boldsymbol{r}, \theta) = h(\boldsymbol{a}^\ell), \\
\boldsymbol{a}^{\ell+1} &= \boldsymbol{a}^\ell.
\end{aligned}
\tag{4}
$$

Here, $h$ is a (graph) neural network that returns a set of equivariant reference vectors $\boldsymbol{r}$, which parametrize the map $\gamma_r$, and invariant parameters $\theta$. $\mathcal{Y}$ is a rotation invariant space. This means that any rotations applied to the inputs will be cancelled by $\gamma_r$, i.e. $\gamma_{g \cdot r} = \gamma_r \cdot g^{-1}$ or equivalently $(\gamma_{g \cdot r})^{-1} = g \cdot \gamma_r^{-1}$ for all $g \in \mathrm{SO}(3)$. We use the inverse projection $\gamma_r^{-1}$ to map the invariant features back to equivariant features. The function $\tau_\theta$ is a standard invertible inner-transformation such as an affine or spline based transform, that we apply to the invariant features [Papamakarios et al., 2021].

We now show that the above described transform is rotation and permutation equivariant.

**Proposition 3.2** (Equivariant augmented coupling flow). *If $h : \mathcal{A} \to \mathcal{X}^n \times \Theta^n$ is $\mathrm{SO}(3)$-equivariant for its first output, $\mathrm{SO}(3)$-invariant for its second, and $S_n$ equivariant for both, and $(\gamma_{g \cdot r})^{-1} = g \cdot \gamma_r^{-1}$ for all $g \in \mathrm{SO}(3)$, the transform $\mathcal{F}$ given by (4) is $\mathrm{SO}(3) \times S_n$ equivariant.*

*Proof.* For $\mathrm{SO}(3)$: We first notice that $h(g \cdot \boldsymbol{a}) = (g \cdot \boldsymbol{r}, \theta)$, and then since $(\gamma_{g \cdot r})^{-1} = g \cdot \gamma_r^{-1}$ we have $\mathcal{F}(g \cdot \boldsymbol{x}, g \cdot \boldsymbol{a}) = (\gamma_{g \cdot r})^{-1} \cdot \tau_\theta(\gamma_{g \cdot r} \cdot g \cdot \boldsymbol{x}^\ell) = g \cdot \gamma_r^{-1} \cdot \tau_\theta(\gamma_r \cdot g^{-1} \cdot g \cdot \boldsymbol{x}^\ell) = g \cdot \mathcal{F}(\boldsymbol{x}, \boldsymbol{a})$.

For $S_n$: We first note that $h(\sigma \cdot \boldsymbol{a}) = (\sigma \cdot \boldsymbol{r}, \sigma \cdot \theta)$. Then, using that $\gamma_r$ and $\tau$ act on $\boldsymbol{x}$ atom-wise, we have $\mathcal{F}(\sigma \cdot \boldsymbol{x}, \sigma \cdot \boldsymbol{a}) = \gamma_{\sigma \cdot r}^{-1} \cdot \tau_{\sigma \cdot \theta}(\gamma_{\sigma \cdot r} \cdot (\sigma \cdot \boldsymbol{x})) = (\sigma \cdot \gamma_r^{-1}) \cdot (\sigma \cdot \tau_\theta)((\sigma \cdot \gamma_r) \cdot (\sigma \cdot \boldsymbol{x})) = \sigma \cdot \mathcal{F}(\boldsymbol{x}, \boldsymbol{a})$. $\square$

For the Jacobian of the coupling described above to be well-defined, the variable being transformed must be non-zero CoM (see App. B.1 for a derivation). Thus, although our observations live on $\tilde{\mathcal{X}}$, for now, assume that the inputs to the transform are not zero-CoM and we will deal with this assumption in the following paragraphs. This choice also allows us to use standard equivariant GNNs for $h$ [Satorras et al., 2021b, Geiger and Smidt, 2022] which leverage per-node features defined in the ambient space, such as atom type and molecular graph connectivity.

**Choices of projection $\gamma$** The equivariant vectors $\boldsymbol{r}$ parameterize a local (per-atom) $\mathrm{SO}(3)$ equivariant reference frame used in the projection $\gamma_r$. We introduce three different projection strategies. (i) The first strategy, is for $\boldsymbol{r}$ to parameterize frame composed of an origin and orthonormal rotation matrix into which we project each atom's positions. We then take $\tau$ to be a dimension-wise transformation for each of the projected atoms' coordinates. We dub this method CARTESIAN-PROJ. (ii) Alternatively, we let $\boldsymbol{r}$ parameterize a origin, zenith direction and azimuth direction for spherical coordinates, as in Liu et al. [2022]. We then apply elementwise transforms to each atom's radius, polar angle and azimuthal angle. We call this SPHERICAL-PROJ. (iii) Lastly, consider a variant of SPHERICAL-PROJ where just the radius is transformed and the polar and azimuth angles are held constant. Here, $\boldsymbol{r}$ parameterizes a single reference point, a per-atom origin. We refer to this last variant as VECTOR-PROJ.

**Architectural details** For the transformations applied in the invariant projected space we consider affine mappings [Dinh et al., 2017] and monotonic rational-quadratic splines [Durkan et al., 2019]. Additionally, to limit computational cost, we have our GNNs $h$ output $M$ sets of reference vectors $\boldsymbol{r}$ and invariant parameters $\theta$. These parametrize $M$ core coupling transformations with a single GNN forward pass. For the CARTESIAN-PROJ and SPHERICAL-PROJ flow we include a loss term that discourages certain reference vectors from being collinear, which improves the projection's stability. We provide further details for this and the various projection types in App. B.3.

**Center of mass shift** The shift-CoM transform allows us to apply the aforementioned $\mathrm{SE}(3) \times S_n$ equivariant coupling in the ambient space rather than zero-COM subspace. In particular, before transforming our observed vector $\tilde{\boldsymbol{x}} \in \tilde{\mathcal{X}}$ with $\mathcal{F}$, we lift it onto $\mathcal{X}$. We achieve this by swapping the center of mass between $\tilde{\boldsymbol{x}}$ and $\boldsymbol{a}$. For now, assume $\mathcal{A} = \mathcal{X}$, i.e. $k = 1$, with App. B.4 providing details for $k > 1$. Letting $\tilde{\mathcal{A}} \subseteq \mathcal{A}$ be the subspace where all augmented variables that differ by

a translation occupy the same point, and $\tilde{a} \in \tilde{\mathcal{A}}$ be defined analogously to $\tilde{x}$, we apply the map $\mathrm{ShiftCoM} : \tilde{\mathcal{X}} \times \mathcal{A} \mapsto \mathcal{X} \times \tilde{\mathcal{A}}$ which acts on both of its arguments by subtracting from each of them the latter's CoM, that is,

$$\mathrm{ShiftCoM}(\tilde{x}, a) \triangleq (\tilde{x} - \bar{a}, \, a - \bar{a}) \quad \text{with} \quad \bar{a} \triangleq \tfrac{1}{n} \sum_{i=1}^{n} [a]^i. \tag{5}$$

This operation is invertible, with inverse $\mathrm{ShiftCoM}(\tilde{a}, x)$, and has unit Jacobian determinant.

| **Algorithm 1:** Flow block $f$ | **Algorithm 2:** Joint density evaluation |
|---|---|
| **Inputs:** Zero-CoM observation $\tilde{x}$, augmented variable $a$, Coupling transforms $\mathcal{F}_1, \mathcal{F}_2$ 

 $(x, \tilde{a}) \leftarrow \mathrm{ShiftCoM}(\tilde{x}, a)$ 
 $(x, \tilde{a}) \leftarrow \mathcal{F}_M^{(1)} \circ \cdots \circ \mathcal{F}_1^{(1)}(x, \tilde{a})$ 
 $(a, \tilde{x}) \leftarrow \mathrm{ShiftCoM}(\tilde{a}, x)$ 
 $(a, \tilde{x}) \leftarrow \mathcal{F}_M^{(2)} \circ \cdots \circ \mathcal{F}_1^{(2)}(a, \tilde{x})$ 

 **Output:** $\tilde{x}, a$ | **Inputs:** $(x, a) \sim p$, base density $q_0$, $(f^{(l)})_{l=1}^L$ 
 $(\tilde{x}^{(0)}, a^{(0)}) \leftarrow (x - \bar{x}, a - \bar{x})$ 
 $\mathrm{logdet} \leftarrow 0$ 
 **for** $l = 1, \dots, L$ **do** 
 $\quad (\tilde{x}^{(l)}, a^{(l)}) \leftarrow f^{(l)}(\tilde{x}^{(l-1)}, a^{(l-1)})$ 
 $\quad \mathrm{logdet} \leftarrow \mathrm{logdet} + \left| \frac{\partial f^{(l)}(\tilde{x}^{(l-1)}, a^{(l-1)})}{\partial (\tilde{x}^{(l-1)}, a^{(l-1)})} \right|$ 

 **Output:** $q_0(\tilde{x}^{(L)}, a^{(L)}) + \mathrm{logdet}$ |

**Putting the building blocks together** Our flow transform is built as a sequence of $L$ blocks. Each block, described in Alg. 1, consists of two equivariant coupling layers, see Fig. 1. Our observations $\tilde{x} \in \tilde{\mathcal{X}}$ are lifted onto $\mathcal{X}$ with $\mathrm{ShiftCoM}$, they are transformed with $M$ core transformations $\left( \mathcal{F}_i^{(1)} \right)_{i=1}^{M}$, and $\mathrm{ShiftCoM}$ is applied one more time to map the observations back to the zero-CoM hyperplane. After this, our augmented variables $a$ are transformed with $\left( \mathcal{F}_i^{(2)} \right)_{i=1}^{M}$.

Joint density evaluation $q(x, a)$ is performed with Alg. 2. We first subtract the observed variables' CoM from both the observed and augmented variables. We then apply our $L$ flow transform blocks before evaluating the transformed variables' density under our base distribution $q_0$, which is described next. The log determinant of the core flow transform, $f$, has a contribution from the projection, transform in the invariant space, and inverse projection (see App. B.3 for details).

## 3.2 SE(3) $\times S_n$ Invariant base distribution

Again, we assume $\mathcal{A} = \mathcal{X}$, i.e. $k = 1$, with the generalization given in App. B.4. Our invariant choice of base distribution is $q_0(x, a) = \tilde{\mathcal{N}}(x; 0, I) \, \mathcal{N}(a; x, \eta^2 I)$ where $x \in \mathbb{R}^{3n}$ and $a \in \mathbb{R}^{3n}$ refer to $x$ and $a$ flattened into vectors, $\eta^2$ is a hyperparameter and we denote Gaussian distributions on $\tilde{\mathcal{X}}$ as $\tilde{\mathcal{N}}$ [Satorras et al., 2021a, Yim et al., 2023] with density

$$\tilde{\mathcal{N}}(x; 0, I) = (2\pi)^{-3(n-1)/2} \exp(-\tfrac{1}{2} \|\tilde{x}\|_2^2). \tag{6}$$

We sample from it by first sampling from a standard Gaussian $\mathcal{N}(0, I)$ and then removing the CoM. On the other hand, the distribution for $a$ is supported on $\mathcal{A}$ which includes non-zero CoM points. It is centered on $x$, yielding joint invariance to translations. The isotropic nature of $q_0$ makes its density invariant to rotations, reflections, and permutations [Satorras et al., 2021a, Yim et al., 2023].

## 3.3 Training and likelihood evaluation

In this section, we discuss learning and density evaluation with augmented variables.

**Invariant augmented target distribution** We assume the density of our observations $p$ is $\mathrm{SE}(3) \times S_n$ invariant. Our target for augmented variables is $\pi(a|x) = \mathcal{N}(a; x, \eta^2 I)$, where $\eta^2$ matches the variance of the base Gaussian density over $a$. This satisfies joint invariance $p(g \cdot x)\pi(g \cdot a|g \cdot x) = p(x)\pi(a|x)$ for any $g \in \mathrm{SE}(3) \times S_n$, as shown in App. B.5.

**Learning from samples** When data samples $x \sim p$ are available, we train our flow parameters by maximizing the joint likelihood, which is a lower bound on the marginal log-likelihood over observations up to a fixed constant

$$\mathbb{E}_{x \sim p(x), a \sim \pi(a|x)}[\log q(x, a)] \leq \mathbb{E}_{p(x)}[\log q(x)] + C. \tag{7}$$

**Learning from energy** When samples are not available but we can query the unnormalized energy of a state $U(x)$, with $p(x) \propto \exp(-U(x))$, we can minimize the joint reverse KL divergence. By

the chain rule of the KL divergence, this upper bounds the KL between marginals

$$D_{\text{KL}}\left(q(\boldsymbol{x},\boldsymbol{a})\,\|\,p(\boldsymbol{x})\pi(\boldsymbol{a}|\boldsymbol{x}))\right) \geq D_{\text{KL}}\left(q(\boldsymbol{x})\,\|\,p(\boldsymbol{x})\right). \tag{8}$$

However, the reverse KL encourages mode-seeking [Minka, 2005] which may result in the model failing to characterize the full set of meta-stable molecular states. Therefore, we instead use *flow annealed importance sampling bootstrap* (FAB) [Midgley et al., 2023], which targets the mass covering $\alpha$-divergence with $\alpha = 2$. In particular, we minimize the $\alpha$-divergence over the joint which leads to an upper bound on the divergence of the marginals

$$D_2\left(q(\boldsymbol{x},\boldsymbol{a})\,\|\,p(\boldsymbol{x})\pi(\boldsymbol{a}|\boldsymbol{x})\right) \triangleq \int \frac{p(\boldsymbol{x})^2\pi(\boldsymbol{a}|\boldsymbol{x})^2}{q(\boldsymbol{x},\boldsymbol{a})}\,\mathrm{d}\boldsymbol{a}\,\mathrm{d}\boldsymbol{x} \geq \int \frac{p(\boldsymbol{x})^2}{q(\boldsymbol{x})}\,\mathrm{d}\boldsymbol{x} \triangleq D_2\left(q(\boldsymbol{x})\,\|\,p(\boldsymbol{x})\right). \tag{9}$$

To compute unbiased expectations with the augmented flow we rely on the estimator $\mathbb{E}_{p(\boldsymbol{x})}[f(\boldsymbol{x})] = \mathbb{E}_{q(\boldsymbol{x},\boldsymbol{a})}[w(\boldsymbol{x},\boldsymbol{a})f(\boldsymbol{x})]$ where $w(\boldsymbol{x},\boldsymbol{a}) = p(\boldsymbol{x})\pi(\boldsymbol{a}|\boldsymbol{x})/q(\boldsymbol{x},\boldsymbol{a})$. Minimizing the joint $\alpha$-divergence with $\alpha = 2$ corresponds to minimizing the variance in the joint importance sampling weights $w(\boldsymbol{x},\boldsymbol{a})$, which allows for the aforementioned expectation to be approximated accurately.

**Evaluating densities** To evaluate the marginal density of observations we use the importance weighed estimator $q(\boldsymbol{x}) = \mathbb{E}_{\boldsymbol{a}\sim\pi(\cdot|\boldsymbol{x})}\left[\frac{q(\boldsymbol{x},\boldsymbol{a})}{\pi(\boldsymbol{a}|\boldsymbol{x})}\right]$, noting that $\pi$ is Gaussian and thus supported everywhere. The estimator variance vanishes when $q(\boldsymbol{a}|\boldsymbol{x}) = \pi(\boldsymbol{a}|\boldsymbol{x})$, as shown in App. B.9.

## 4 Experiments

### 4.1 Training with samples: DW4, LJ13 and QM9 positional

First, we consider 3 problems that involve only positional information, with no additional features such as atom type or connectivity. Thus, the target densities are fully permutation invariant. The first two of these, namely DW4 and LJ13, are toy problems from Köhler et al. [2020], where samples are obtained by running MCMC on the 4 particle double well energy function (DW4) and 13 particles Leonard Jones energy function (LJ13) respectively. The third problem, i.e. QM9 positional [Satorras et al., 2021a], selects the subset of molecules with 19 atoms from the commonly used QM9 dataset [Ramakrishnan et al., 2014] and discards their node features.

For our model, Equivariant Augmented Coupling Flow (E-ACF), we consider all projection types (VECTOR-PROJ, CARTESIAN-PROJ, SPHERICAL-PROJ) and compare them to: (1) NON-E-ACF: An augmented flow that is not rotation equivariant but is translation and permutation equivariant, as in [Klein et al., 2023a]. This model uses the same structure as the E-ACF but replaces the EGNN with a transformer which acts directly on atom positions, without any projection. We train NON-E-ACF with data-augmentation whereby we apply a random rotation to each sample within each training batch. (2) E-CNF ML: The SE(3) equivariant continuous normalizing flow from Satorras et al. [2021a] trained by maximum likelihood. (3) E-CNF FM: An SE(3) equivariant continuous normalizing flow trained via flow matching [Lipman et al., 2023, Klein et al., 2023b]. (4) E-CNF-DIFF: An SE(3) equivariant diffusion model [Hoogeboom et al., 2022] evaluated as a continuous normalizing flow. All equivariant generative models use the SE(3) GNN architecture proposed by Satorras et al. [2021b]. The CARTESIAN-PROJ exhibited numerical instability on QM9-positional causing runs to crash early. To prevent these crashes it was trained at a lower learning rate than the other E-ACF models. App. C.3.1 provides a detailed description of these experiments.

On Tab. 1, we see that on DW4 and LJ13, E-CNF FM performs best, while E-ACF is competitive with E-CNF ML and E-CNF-DIFF. We note that both DW4 and LJ13 have biased datasets (see App. C.3.1). On QM9-positional, the E-ACF is competitive with E-CNF-DIFF and E-CNF FM, while the under-trained E-CNF ML from Satorras et al. [2021a] performs poorly. We expect that with further tuning E-CNF-DIFF and E-CNF FM would match the performance of SPHERICAL-PROJ on QM9-positional. The NON-E-ACF performs much worse than the E-ACF, despite being trained for more epochs, demonstrating the utility of in-build equivariance. Furthermore, Fig. 2 shows that the distribution of inter-atomic distances of samples from our flow matches training data well. Importantly, sampling and density evaluation of the E-ACF on an A100 GPU takes roughly 0.01 seconds. For the CNF trained by flow matching (E-CNF ML) and score matching (E-CNF-DIFF), sampling takes on average 0.2, and 5 seconds respectively. Thus, the E-ACF is faster for sampling than the CNF by more than an order of magnitude.

Table 1: Negative log-likelihood results for flows trained by maximum likelihood on DW4, LJ13 and QM9-positional. E-CNF ML results are from Satorras et al. [2021a]. Best results are emphasized in **bold**. The results are averaged over 3 seeded runs, with the standard error reported as uncertainty.

|  | DW4 | LJ13 | QM9 positional |
|---|---|---|---|
| E-CNF ML | $8.15 \pm N/A$ | $30.56 \pm N/A$ | $-70.2 \pm N/A$ |
| E-CNF FM | $\mathbf{6.31 \pm 0.05}$ | $\mathbf{28.81 \pm 0.34}$ | $-143.52 \pm 0.26$ |
| E-CNF-Diff | $8.01 \pm 0.03$ | $31.02 \pm 0.12$ | $-158.30 \pm 0.15$ |
| Non-E-ACF | $10.07 \pm 0.03$ | $33.32 \pm 0.34$ | $-76.76 \pm 1.77$ |
| Vector-proj E-ACF | $8.69 \pm 0.03$ | $30.19 \pm 0.12$ | $-152.23 \pm 6.44$ |
| Cartesian-proj E-ACF | $8.82 \pm 0.08$ | $30.89 \pm 0.09$ | $-138.62 \pm 0.74$ |
| Spherical-proj E-ACF | $8.61 \pm 0.05$ | $30.33 \pm 0.16$ | $\mathbf{-165.71 \pm 1.35}$ |

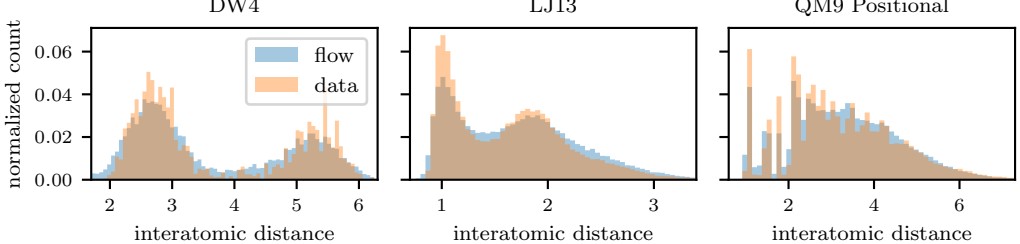

Figure 2: Inter-atomic distances for samples from the train-data and SPHERICAL-PROJ E-ACF.

## 4.2 Training with samples: Alanine dipeptide

Next, we approximate the Boltzmann distribution of alanine dipeptide in an implicit solvent at temperature $T = 800 \, \text{K}$. We train the models via maximum likelihood on samples generated by a replica exchange MD simulation [Mori and Okamoto, 2010], which serve as a ground truth. Our generated dataset consists of $10^6$ samples in the training and validation set as well as $10^7$ samples in the test set. Besides our E-ACF using the different projection schemes that we introduced in Sec. 3.1, we train the non-SO(3) equivariant flow (NON-E-ACF) with data augmentation similarly to the previous experiments. Moreover, we train a flow on internal coordinates as in Midgley et al. [2023]. The joint effective sample size (ESS) is reported for the E-ACF models which is a lower bound of the marginal ESS (see Eq. (9)). For the CNFs, the Hutchinson trace estimator is used for the log density [Grathwohl et al., 2018]. This results in a biased estimate of the ESS, which may therefore be spurious. Further details about model architectures, training and evaluation are given in App. C.3.2.

Table 2: Alanine dipeptide results. KLD is the empirical KLD of the Ramachandran plots (see Fig. 3). Forward ESS is estimated with the test set. Reverse ESS is estimated with $10^5$ model samples. Results are averaged over 3 seeded runs, with the standard error reported as uncertainty.

|  | KLD | NLL | Rev ESS (%) | Fwd ESS (%) |
|---|---|---|---|---|
| Flow on internal coordinates | $(2.01 \pm 0.04) \cdot 10^{-3}$ | $-190.15 \pm 0.02$ | $1.61 \pm 0.03$ | $-$ |
| E-CNF FM | $(3.83 \pm 0.18) \cdot 10^{-2}$ | $\mathbf{-190.20 \pm 0.09}$ | $(3.1 \pm 0.2) \cdot 10^{-4}$ | $(2.5 \pm 0.8) \cdot 10^{-128}$ |
| E-CNF-Diff | $(8.86 \pm 0.49) \cdot 10^{-3}$ | $-188.31 \pm 0.01$ | $(8.1 \pm 1.1) \cdot 10^{-4}$ | $(5.1 \pm 4.1) \cdot 10^{-237}$ |
| Non-E-ACF | $(1.66 \pm 0.01) \cdot 10^{-1}$ | $-184.57 \pm 0.35$ | $0.14 \pm 0.07$ | $(5.5 \pm 4.5) \cdot 10^{-30}$ |
| Vector-proj E-ACF | $(6.15 \pm 1.21) \cdot 10^{-3}$ | $-188.56 \pm 0.01$ | $19.4 \pm 13.4$ | $(\mathbf{9.4 \pm 7.7}) \cdot \mathbf{10^{-7}}$ |
| Cartesian-proj E-ACF | $(3.46 \pm 0.28) \cdot 10^{-3}$ | $-188.59 \pm 0.00$ | $\mathbf{52.5 \pm 3.2}$ | $(9.7 \pm 7.9) \cdot 10^{-9}$ |
| Spherical-proj E-ACF | $(\mathbf{2.55 \pm 0.29}) \cdot \mathbf{10^{-3}}$ | $-188.57 \pm 0.00$ | $48.4 \pm 7.2$ | $(5.0 \pm 4.1) \cdot 10^{-14}$ |

The results are shown in Fig. 3 and Tab. 2. All variants of our E-ACF clearly outperform the NON-E-ACF, as the Kullback Leibler divergence (KLD) of the Ramachandran plots is significantly lower and the NLL as well as the reverse and forward ESS, see App. B.10, are higher. The flow trained on internal coordinates is only marginally better regarding the NLL and KLD than our best models, i.e.

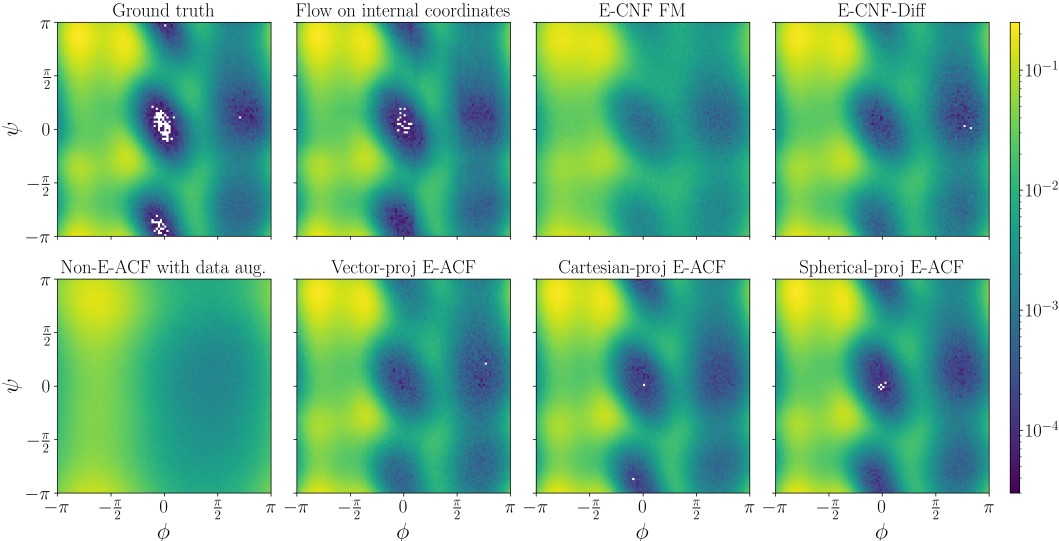

Figure 3: Ramachandran plots, i.e. marginal distribution of the dihedral angles $\phi$ and $\psi$ (see App. C.3.2), obtained with MD (ground truth) and various models.

the CARTESIAN-PROJ and SPHERICAL-PROJ E-ACF, while we outperform it considering the ESS. Note that the flow on internal coordinates explicitly models the angles $\phi$ and $\psi$, while the E-ACF operates on the underlying Cartesian coordinates. The E-ACFs outperform the E-CNF-DIFF model in all metrics, but the E-CNF trained with flow matching has a slightly lower NLL while having a significantly higher KLD on the Ramachandran plot. This could be due to a better performance on other marginals or on some outlier data points in the test set. The forward ESS is very low for all models, which suggests that the models do not cover some regions in the target distribution, and that the reverse ESS is spurious. Alternatively, this may be from numerical instabilities in the models. To the best of our knowledge, our models are the first to learn the full Boltzmann distribution of a molecule purely on Cartesian coordinates while being competitive with a flow trained on internal coordinates. Additionally, we train the best E-ACF model at $T = 300\,\mathrm{K}$ where it also performs well (see App. C.3.2).

## 4.3 Energy-based training: DW4 and LJ13

Lastly, we demonstrate that our proposed flow can be trained on the DW4 and LJ13 problems using only the target's unnormalized density with the FAB algorithm [Midgley et al., 2023]. The annealed importance sampling procedure within FAB requires sampling from the flow and evaluating its density multiple times. This is used within the training loop of FAB making it significantly more expensive per parameter update than training by maximum likelihood. Given that sampling and density evaluation with CNFs is very expensive, training them with FAB is intractable. Thus, we only report results for our flow, as well as for the NON-E-ACF. We train the NON-E-ACF for more iterations than the E-ACF, such that the training times are similar, given that the NON-E-ACF is faster per iteration. App. C.4 provides further details on the experimental setup.

Table 3: Results for training by energy with FAB. Best results are emphasized in **bold**. Results are averaged over 3 seeded runs, with the standard error reported as uncertainty. Reverse ESS is estimated with $10^4$ samples from the flow. Forward ESS is estimated with the test sets.

| | DW4 | | | LJ13 | | |
|---|---|---|---|---|---|---|
| | Rev ESS (%) | Fwd ESS (%) | NLL | Rev ESS (%) | Fwd ESS (%) | NLL |
| NON-E-ACF | $35.94 \pm 2.63$ | $5.45 \pm 4.10$ | $7.38 \pm 0.01$ | $5.38 \pm 3.66$ | $4.14 \pm 3.10$ | $33.22 \pm 0.96$ |
| VECTOR-PROJ E-ACF | $\mathbf{84.29 \pm 0.41}$ | $\mathbf{83.39 \pm 0.79}$ | $\mathbf{7.11 \pm 0.00}$ | $59.60 \pm 1.13$ | $65.20 \pm 1.61$ | $30.33 \pm 0.03$ |
| CARTESIAN-PROJ E-ACF | $82.44 \pm 0.50$ | $80.08 \pm 0.64$ | $7.13 \pm 0.00$ | $60.68 \pm 0.41$ | $65.54 \pm 0.37$ | $30.34 \pm 0.01$ |
| SPHERICAL-PROJ E-ACF | $80.44 \pm 0.88$ | $81.46 \pm 0.95$ | $7.14 \pm 0.00$ | $\mathbf{62.09 \pm 0.76}$ | $\mathbf{66.13 \pm 0.11}$ | $\mathbf{30.21 \pm 0.02}$ |

Tab. 3 shows that the E-ACF trained with FAB successfully approximates the target Boltzmann distributions, with reasonably high joint ESS, and NLL comparable to the flows trained by maximum likelihood. Additionally, the ESS may be improved further by combining the trained flow with AIS, this is shown in App. C.4. In both problems the NON-E-ACF performs worse, both in terms of ESS and NLL. All models trained by maximum likelihood have a much lower ESS (see App. C.3.1). This is expected, as unlike the $\alpha = 2$-divergence loss used in FAB, the maximum likelihood objective does not explicitly encourage minimizing importance weight variance. Furthermore, the flows trained by maximum likelihood use a relatively small, biased training set, which therefore limits their quality.

## 5 Discussion and Related Work

Augmented flows have been used for improving expressiveness of Boltzmann generators [Köhler et al., 2023b,c]; however, these models were not equivariant. Klein et al. [2023a] proposed an augmented normalizing flow architecture to provide conditional proposal distributions for MD simulations and use a coupling scheme similar to ours. However, this model only achieves translation and permutation equivariance and the authors make their flow approximately rotation invariant through data augmentation. In our experiments, we found data augmentation to perform significantly worse than intrinsic invariance. In principle, Klein et al. [2023a]'s model could be made fully invariant by substituting in our flow's projection-based coupling transform.

An alternative to our equivariant flow and equivariant CNFs are the equivariant residual flows proposed in Bose et al. [2022]. Alas, residual flows require fixed-point iteration for training and inversion. This is expensive and may interact poorly with energy-based training methods such as FAB [Midgley et al., 2023] which require fast exact sampling and densities. Furthermore, Bose et al. [2022] found that the spectral normalization required for residual flows did not interact well with the equivariant CNN architecture in their experiments.

There has been recent progress in improving the sampling speed of CNFs/diffusion models [Tong et al., 2023, Song et al., 2023] and on using these models for sampling from unnormalized densities [Vargas et al., 2023, Zhang and Chen, 2022, Zhang et al., 2023]. Thus, in the future CNF/diffusion models trained by energy may prove to be competitive with discrete-time flow based methods.

The strategy of projection into a local reference frame to enforce equivariance has been successfully employed in existing literature, specifically for protein backbone generation [Jumper et al., 2021, Yim et al., 2023]. Here we have focused on modelling the full set of Cartesian coordinates of a molecule, but an interesting avenue for future work is to extend our framework to other domains, such as modelling rigid bodies, which has applications to protein backbone generation [Jumper et al., 2021, Yim et al., 2023] and many-body systems [Köhler et al., 2023c].

**Limitations** Although our flow is significantly faster than alternatives such as CNFs, the expensive EGNN forward pass required in each layer of the flow makes it more computationally expensive than flows on internal coordinates. Additionally, we found our flow to be less numerically stable than flows on internal coordinates, which we mitigate via adjustments to the loss, optimizer and neural network (see App. B.3, App. C.1, App. C.2). Our implementation uses the E(3) equivariant EGNN proposed by Satorras et al. [2021a]. However, recently there have been large efforts towards developing more expressive, efficient and stable EGNNs architectures [Fuchs et al., 2020, Batatia et al., 2022, Musaelian et al., 2023, Liao and Smidt, 2023]. Incorporating these into our flow may improve performance, efficiency and stability. This would be especially useful for energy-based training, where the efficiency of the flow is a critical factor.

## 6 Conclusion

We have proposed an SE(3) equivariant augmented coupling flow that achieves similar performance to CNFs when trained by maximum likelihood, while allowing for faster sampling and density evaluation by more than an order of magnitude. Furthermore, we showed that our flow can be trained as a Boltzmann generator using only the target's unnormalized density, on problems where internal coordinates are inadequate due to permutation symmetries, and doing so with a CNF is computationally intractable. It is possible to extend our model to learn the Boltzmann distribution of diverse molecules, by conditioning on their molecular graph, which we hope to explore in the future.

## Acknowledgments and Disclosure of Funding

We thank Gábor Csányi and his group for the helpful discussions. Laurence Midgley acknowledges support from Google's TPU Research Cloud (TRC) program and from the EPSRC through the Syntech PhD program. Vincent Stimper acknowledges the Max Planck Computing and Data Facilities for providing extensive computing resources and support. Javier Antorán acknowledges support from Microsoft Research, through its PhD Scholarship Programme, and from the EPSRC. José Miguel Hernández-Lobato acknowledges support from a Turing AI Fellowship under grant EP/V023756/1. José Miguel Hernández-Lobato and Emile Mathieu are supported by an EPSRC Prosperity Partnership EP/T005386/1 between Microsoft Research and the University of Cambridge. This work has been performed using resources provided by the Cambridge Tier-2 system operated by the University of Cambridge Research Computing Service (http://www.hpc.cam.ac.uk) funded by an EPSRC Tier-2 capital grant. It was also supported by the German Federal Ministry of Education and Research (BMBF): Tübingen AI Center, FKZ: 01IS18039B; and by the Machine Learning Cluster of Excellence, EXC number 2064/1 - Project number 390727645.

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

# A  Background

## A.1  Tensors and representation

We assume a group $(G, \cdot)$, an algebraic structure that consists of a set $G$ with a group operator $\cdot$ that follows the *closure*, *inverse* and *associativity* axioms.

A representation is an invertible linear transformation $\rho(g) : V \to V$, indexed by group elements $g \in G$, and defines how group elements $g$ acts on elements of the vector space $V$. It satisfies the group structure $\rho(gh) = \rho(g)\rho(h)$.

A *tensor* $T$ is a geometrical object that is acted on by group elements $g \in G$ in a particular way, characterized by its representation $\rho$: $g \cdot T = \rho(g)T$. Typical examples include scalar with $\rho_{\mathrm{triv}}(g) = 1$ and vectors living in $V$ with $\rho_{\mathrm{Id}}(g) = g$. Higher order tensors which live in $V \times \cdots \times V$ can be constructed as $g \cdot T = (\rho_{\mathrm{Id}}(g) \otimes \cdots \otimes \rho_{\mathrm{Id}}(g))\,\mathrm{vec}(T)$. A collection of tensors $(T_1, \ldots, T_n)$ with representations $(\rho_1, \ldots, \rho_n)$ can be stacked with fiber representation as the direct sum $\rho = \rho_1 \oplus \cdots \oplus \rho_n$.

## A.2  Translational invariant densities

**Proposition A.1** (Disintegration of measures for translation invariance [Pollard, 2002, Yim et al., 2023])**.** *Under mild assumptions, for every* $\mathrm{SE}(3)$*-invariant measure* $\mu$ *on* $\mathbb{R}^{3 \times n}$*, there exists* $\eta$ *an* $\mathrm{SO}(3)$*-invariant probability measure on* $\{\mathbb{R}^{3 \times n} | \sum_{i=1}^{n} x_i = 0\}$ *and* $\bar{\mu}$ *proportional to the Lebesgue measure on* $\mathbb{R}^3$ *s.t.* $\mathrm{d}\mu([x_1, \ldots, x_n]) = \mathrm{d}\bar{\mu}(\frac{1}{n}\sum_{i=1}^{n} x_i) \times \mathrm{d}\eta([x_1 - \frac{1}{n}\sum_{i=1}^{n} x_i, \ldots, x_n - \frac{1}{n}\sum_{i=1}^{n} x_i])$.

The converse is also true, which motivates constructing an $\mathrm{SO}(3)$-invariant probability measure on the subspace of $\mathbb{R}^{3 \times n}$ with zero center of mass.

# B  Method

## B.1  Transforming variables on a subspace

This section shows that a flow transform $\mathcal{F} : \mathcal{X} \to \mathcal{X}$ applied to observations $\tilde{x}$ embedded in the zero-CoM subspace $\tilde{\mathcal{X}}$ does not have a well-defined Jacobian. We will show this by constructing such a transform, which leaves zero-CoM input unchanged, but for which the log-determinant is arbitrary.

Without loss of generality, we will assume in this subsection and the next that our observed and augmented variables are one dimensional, i.e., $\mathcal{X} = \mathcal{A} = \mathbb{R}^n$. Consider the orthonormal basis $V \in \mathbb{R}^{n \times n}$ constructed by concatenating orthonormal basis vectors $v_i \in \mathbb{R}^n$ as $[v_1, \ldots, v_n]^T$. We choose $v_1$ to be equal to $[1, 1, \ldots, 1]^T / \sqrt{n}$ such that inner products with $v_1$ retrieve the CoM of a system of particles up to multiplication by a scalar: $v_1^T x = \sqrt{n}\bar{x}$ and $v_1^T \tilde{x} = 0$. Let $\mathcal{F}(x) = V^T S V x$ be a mapping which rotates onto the basis defined by $V$, multiplies by the diagonal matrix $S = \mathrm{diag}([s, 1, \ldots, 1])$ and then rotates back to the original space with $V^T$. This transformation only acts on the CoM. Applying this transformation to a zero-CoM variable $\tilde{x}$ we have

$$\mathcal{F}(\tilde{x}) = V^T S V \tilde{x} = V^T [s v_1^T \tilde{x}, v_2^T \tilde{x}, \ldots, v_n^T \tilde{x}] = V^T [0, v_2^T \tilde{x}, \ldots, v_n^T \tilde{x}] = \tilde{x},$$

leaving the zero CoM input unchanged. However, the log-determinant is

$$\log \left| \frac{\partial \mathcal{F}(x)}{\partial x} \right| = \log \left| V^T S V \right| = \log |V^T| + \log |S| + \log |V| = \log s,$$

since for the unitary matrices $|V| = |V^T| = 1$. Because $s$ is arbitrary, so is the density returned by any flow applying such a layer to zero-CoM variables. In the maximum likelihood setting, the aforementioned flow would lead to a degenerate solution of $s \to 0$ such that the log determinant would approach negative infinity and the log likelihood would approach infinity.

While our flow does use transforms which operate on non-CoM variables, we lift our CoM observations to the non-zero-CoM space before applying said transformations. We do this using the ShiftCoM operation, described in Sec. 3 and below.

## B.2 Jacobian of ShiftCoM

We now show our $\text{ShiftCoM} : \tilde{\mathcal{X}} \times \mathcal{A} \mapsto \mathcal{X} \times \tilde{\mathcal{A}}$ transformation to have unit Jacobian log determinant. We first, re-state the function definition

$$\text{ShiftCoM}(\tilde{\boldsymbol{x}}, \boldsymbol{a}) \triangleq (\tilde{\boldsymbol{x}} - \bar{\boldsymbol{a}}, \, \boldsymbol{a} - \bar{\boldsymbol{a}}) \quad \text{with} \quad \bar{\boldsymbol{a}} \triangleq \frac{1}{n} \sum_{i=1}^{n} [\boldsymbol{a}]^i.$$

We now re-write it using the orthonormal basis $V \in \mathbb{R}^{n \times n}$ defined in the previous subsection. For this, we use $I_2 \otimes V \in \mathbb{R}^{2n \times 2n}$ to refer to a $2 \times 2$ block diagonal matrix with $V$ in both blocks, and we also stack the inputs and outputs of ShiftCoM, yielding $\text{ShiftCoM}([\tilde{\boldsymbol{x}}, \boldsymbol{a}]) = (I_2 \otimes V)^T P (I_2 \otimes V)[\tilde{\boldsymbol{x}}, \boldsymbol{a}]$, where $P$ is a permutation matrix that exchanges the first and $n+1$th elements of the vector it acts upon. It also flips this element's sign. To see this note

$$\begin{aligned}
\text{ShiftCoM}([\tilde{\boldsymbol{x}}, \boldsymbol{a}]) &= (I_2 \otimes V)^T P (I_2 \otimes V)[\tilde{\boldsymbol{x}}, \boldsymbol{a}] \\
&= (I_2 \otimes V)^T P [0, \boldsymbol{v}_2^T \tilde{\boldsymbol{x}}, \dots, \boldsymbol{v}_n^T \tilde{\boldsymbol{x}}, \boldsymbol{v}_1^T \boldsymbol{a}, \dots, \boldsymbol{v}_n^T \boldsymbol{a}] \\
&= (I_2 \otimes V)^T [-\boldsymbol{v}_1^T \boldsymbol{a}, \boldsymbol{v}_2^T \tilde{\boldsymbol{x}}, \dots, \boldsymbol{v}_n^T \tilde{\boldsymbol{x}}, 0, \boldsymbol{v}_2^T \boldsymbol{a}, \dots, \boldsymbol{v}_n^T \boldsymbol{a}] \\
&= [\tilde{\boldsymbol{x}} - \bar{\boldsymbol{a}}, \boldsymbol{a} - \bar{\boldsymbol{a}}].
\end{aligned}$$

For the determinant we use that $|I_2| = 1$ and $|V| = 1$, and thus the determinant of their Kronecker product will be equal to one $|(I_2 \otimes V)| = 1$. Combining this with the fact that permutation matrices and matrices that flip signs have unit determinant, we arrive at

$$\begin{aligned}
\log \left| \frac{\partial \text{ShiftCoM}([\tilde{\boldsymbol{x}}, \boldsymbol{a}])}{\partial [\tilde{\boldsymbol{x}}, \boldsymbol{a}]} \right| &= \log \left| (I_2 \otimes V)^T P (I_2 \otimes V) \right| \\
&= \log |(I_2 \otimes V)^T| + \log |P| + \log |I_2 \otimes V| = 1.
\end{aligned}$$

It is worth noting that ShiftCoM preserves the inherent dimensionality of its input; both its inputs and outputs consist of one zero-CoM vector and one non-zero-CoM vector. Because ShiftCoM does not apply any transformations to the CoM, it does not suffer from the issue of ill-definedness discussed in the previous subsection.

## B.3 Projection variants and their respective transformations

In this section we provide a detailed description of the core transformation of our equivariant coupling flow layer, see Fig. 1. It consists of a projection into an invariant space via an equivariant reference, followed by a flow transformation in this space and a back-projection.

We denote the inputs with superscript $\ell$ and use $\ell + 1$ for outputs to define our transformation block $\mathcal{F} : (\boldsymbol{x}^\ell; \boldsymbol{a}^\ell) \mapsto (\boldsymbol{x}^{\ell+1}, \boldsymbol{a}^{\ell+1})$. Without loss of generality, we assume that we are transforming $\boldsymbol{x}^\ell$. Then, $\mathcal{F}$ takes the following form.

$$\begin{aligned}
\boldsymbol{x}^{\ell+1} &= \gamma_{\boldsymbol{r}}^{-1} \cdot \tau_\theta(\gamma_{\boldsymbol{r}} \cdot \boldsymbol{x}^\ell), \quad \text{with } (\boldsymbol{r}, \theta) = h(\boldsymbol{a}^\ell), \\
\boldsymbol{a}^{\ell+1} &= \boldsymbol{a}^\ell.
\end{aligned} \tag{10}$$

Here, $h$ is a (graph) neural network that returns a set of $u$ equivariant reference vectors $\boldsymbol{r} = [r^1, \dots, r^u]$, and invariant parameters $\theta$. The equivariant vectors $\boldsymbol{r}$ parametrize $\gamma_{\boldsymbol{r}}$, a projection operator onto a rotation invariant feature space. This means that any rotations applied to the inputs will be cancelled by $\gamma_{\boldsymbol{r}}$, i.e. $\gamma_{g \cdot \boldsymbol{r}} = \gamma_{\boldsymbol{r}}^{-1} \cdot g^{-1}$ or equivalently $(\gamma_{g \cdot \boldsymbol{r}})^{-1} = g \cdot \gamma_{\boldsymbol{r}}^{-1}$ for all $g \in \text{SO}(3)$.

In the next paragraphs, we discuss three projection operations and their respective flow transformations.

**Cartesian projection** Here, we predict a 3D Cartesian coordinate system as illustrated in Fig. 4a. The coordinate system has an origin $o = r^1$ and an orthonormal basis $Q$, both being equivariant. $Q$ is constructed from two equivariant vectors $v_1 = r^2 - r^1$ and $v_2 = r^3 - r^1$ in the following way. To obtain the first basis vector, we simply normalize $v_1$, i.e. $b_1 = \frac{v_1}{\|v_1\|}$. We get the second basis vector via a Gram-Schmidt [Petersen, 2012] iteration, i.e. we compute the component of $v_2$ orthogonal to $v_1$, which is $v_{2,\perp} = v_2 - b_1^\top v_2$, and then normalize it to obtain $b_2 = \frac{v_{2,\perp}}{\|v_{2,\perp}\|}$. Note that this is ill-posed if $v_1$ and $v_2$ are collinear. To avoid this case, we introduce an auxiliary loss,

$$\text{aux-loss} = -\log\left(\epsilon + \arccos(\text{abs}(v_1^T v_2 / \|v_1\| / \|v_2\|))\right), \tag{11}$$

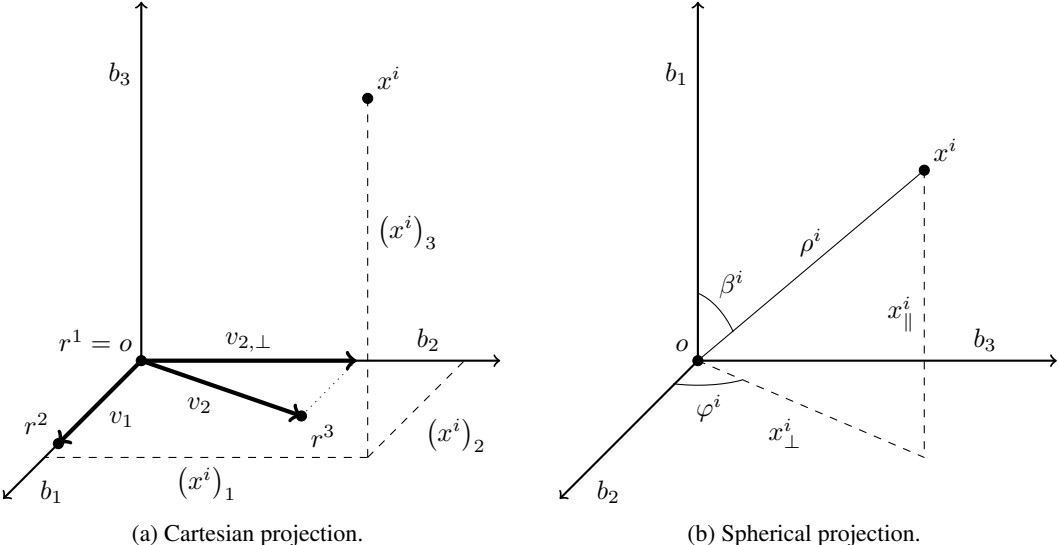

(a) Cartesian projection.      (b) Spherical projection.

Figure 4: Illustration of the Cartesian and spherical projection.

which uses a logarithmic barrier function on the angle between the vectors to prevent them from being collinear. $\epsilon$ is a small constant that prevents the aux-loss from being equal to infinity. Finally, the third basis vector is $b_3 = b_1 \times b_2$, and we can assemble $Q = [b_1, b_2, b_3]$.

The projection into this coordinate system and the respective inverse operation are given by

$$\gamma_{\boldsymbol{r}}(\boldsymbol{x}) = Q^{-1}(\boldsymbol{x} - o)), \tag{12}$$

$$\gamma_{\boldsymbol{r}}^{-1}(\boldsymbol{x}_{\mathrm{p}}) = Q\boldsymbol{x}_{\mathrm{p}} + o, \tag{13}$$

where $\boldsymbol{x}_{\mathrm{p}}$ is the transformed projected variable.

In the projected space, we can apply any standard flow transform, as long as it acts on each particle individually. We choose $\tau_\theta$ to an affine transformation as in the Real NVP architecture [Dinh et al., 2017].

Note that since $b_3$ is constructed via a cross product, it is not parity equivariant, which is why the entire transformation block is not parity equivariant either. However, since $\mathrm{sign}(b_3^\top r^2)$ is a pseudo-scalar, we can replace $b_3$ with $b_3' = \mathrm{sign}(b_3^\top r^2)b_3$, which is a vector, to retain parity equivariance.

The determinant of the Jacobian of $\mathcal{F}$ when using the Cartesian projection is just the determinant of the Jacobian of $\tau_\theta$, which we show below.

$$\left| \frac{\partial \mathcal{F}(\boldsymbol{x}; \boldsymbol{a})}{\partial(\boldsymbol{x}, \boldsymbol{a})} \right| = \left| \frac{\partial \mathcal{F}(\boldsymbol{x}; \boldsymbol{a})|_{\boldsymbol{x}}}{\partial \boldsymbol{x}} \right| \cdot \left| \frac{\partial \mathcal{F}(\boldsymbol{x}; \boldsymbol{a})|_{\boldsymbol{a}}}{\partial \boldsymbol{a}} \right|$$

$$= \left| \left[ \frac{\partial \gamma_{\boldsymbol{r}}^{-1}(\boldsymbol{z})}{\partial \boldsymbol{z}} \right]_{\boldsymbol{z}=\tau_\theta(\gamma_{\boldsymbol{r}}(\boldsymbol{x}))} \left[ \frac{\partial \tau_\theta(\boldsymbol{z})}{\partial \boldsymbol{z}} \right]_{\boldsymbol{z}=\gamma_{\boldsymbol{r}}(\boldsymbol{x})} \frac{\partial \gamma_{\boldsymbol{r}}(\boldsymbol{x})}{\partial \boldsymbol{x}} \right| \cdot 1$$

$$= \left| Q \left[ \frac{\partial \tau_\theta(\boldsymbol{z})}{\partial \boldsymbol{z}} \right]_{\boldsymbol{z}=\gamma_{\boldsymbol{r}}(\boldsymbol{x})} Q^{-1} \right|$$

$$= |Q| \left| \left[ \frac{\partial \tau_\theta(\boldsymbol{z})}{\partial \boldsymbol{z}} \right]_{\boldsymbol{z}=\gamma_{\boldsymbol{r}}(\boldsymbol{x})} \right| |Q^{-1}|$$

$$= \left| \left[ \frac{\partial \tau_\theta(\boldsymbol{z})}{\partial \boldsymbol{z}} \right]_{\boldsymbol{z}=\gamma_{\boldsymbol{r}}(\boldsymbol{x})} \right|.$$

**Spherical projection** For the spherical projection, which is illustrated in Fig. 4b, we compute an origin $o$ and three Cartesian basis vectors $b_1$, $b_2$, and $b_3$ as in the Cartesian projection. Instead of

computing the Cartesian coordinates of each $x^i$, we compute the spherical coordinates $(\rho, \beta, \varphi)$. Hence, for the $i$-th atom the projection operation and its inverse is given by

$$(\gamma_{\boldsymbol{r}}(\boldsymbol{x}))^i = \left( \|x^i\|, \arccos\left(\frac{x^i_\parallel}{\|x^i\|}\right), \operatorname{atan2}\left(b_3^\top x^i_\perp, b_2^\top x^i_\perp\right) \right)^\top, \tag{14}$$

$$\left(\gamma_{\boldsymbol{r}}^{-1}(\boldsymbol{x}_{\mathrm{p}})\right)^i = \rho^i \left(\cos(\beta^i)b_1 + \sin(\beta^i)\left(\cos(\varphi^i)b_2 + \sin(\varphi^i)b_3\right)\right), \tag{15}$$

Here, we used $x^i_\parallel = b_1^\top x^i$, $x^i_\perp = x^i - x^i_\parallel$, and $x^i_{\mathrm{p}} = (\rho^i, \beta^i, \varphi^i)$ for brevity.

The spherical coordinates are transformed with monotonic rational-quadratic splines [Durkan et al., 2019]. The radial coordinate $\rho$ is transformed through a spline with the interval starting at 0 to ensure that the transformed coordinate is also non-negative to ensure invertibility. The two angles are transformed via circular splines [Rezende et al., 2020]. Similarly to the Cartesian projection, we use the auxiliary loss (Eq. (11)) to prevent the vectors output from the GNN from being co-linear.

When using the spherical projection, we compute the log-determinant of the Jacobian of $\mathcal{F}$ by using the chain rule in the same way as we do for standard normalizing flows. The Jacobian determinant of the projection operation is given by the well-known term

$$\left| \frac{\partial \left(\gamma_{\boldsymbol{r}}(\boldsymbol{x})\right)^i}{\partial \boldsymbol{x}} \right| = \left(\rho^i\right)^2 \sin \beta^i. \tag{16}$$

For the inverse, we can just raise this term to the power of $-1$.

**Vector projection** This is a special case of the spherical projection, where we only transform the radial coordinate but leave the angles the same. Since in practice we only need to compute $\rho$, we call this case vector projection. Note that just predicting the origin $o$ as a reference with the equivariant neural network is sufficient here. Unlike the Cartesian and spherical projections, the vector projection does not rely on the cross product, and hence is parity equivariant by default.

## B.4  Multiple augmented variables

Consider augmented variables consisting of multiple sets of observation-sized arrays, i.e. $\mathcal{A} = \mathcal{X}^k$ with $k > 1$. For the base distribution, and also the target, we deal with multiple augmented variables by letting them follow conditional normal distributions centered at $\boldsymbol{x}$. We thus have $\pi(\boldsymbol{a}|\boldsymbol{x}) = \prod_{i=1}^k \mathcal{N}(\boldsymbol{a}_i; \boldsymbol{x}, \sigma^2 I)$ and $q_0(\boldsymbol{x}, \boldsymbol{a}) = \tilde{\mathcal{N}}(\boldsymbol{x}; 0, I) \prod_{i=1}^k \mathcal{N}(\boldsymbol{a}_i; \boldsymbol{x}, \sigma^2 I)$.

For our flow block, we group our variables into two sets $(\tilde{\boldsymbol{x}}, \boldsymbol{a}_1, \dots, \boldsymbol{a}_{(k-1)/2})$ and $(\boldsymbol{a}_{(k-1)/2+1}, \dots, \boldsymbol{a}_k)$, and update one set conditional on the other, and vice-versa, while always ensuring the arrays being transformed have non-zero CoM.

Initially, $\boldsymbol{a}$ is non-zero-CoM with each augmented variable $\boldsymbol{a}_i$ having a CoM located at a different point in space. Before the first coupling transform $\mathcal{F}_M^{(1)}$, the $\mathrm{ShiftCoM}$ function applies the mapping

$$\mathrm{ShiftCoM}(\tilde{\boldsymbol{x}}, \boldsymbol{a}) \triangleq (\tilde{\boldsymbol{x}} - \bar{\boldsymbol{a}}_{1+\frac{k-1}{2}}, \boldsymbol{a} - \bar{\boldsymbol{a}}_{1+\frac{k-1}{2}}) \quad \text{with} \quad \bar{\boldsymbol{a}}_{1+\frac{k-1}{2}} \triangleq \frac{1}{n}\sum_{i=1}^n [\boldsymbol{a}_{1+\frac{k-1}{2}}]^i. \tag{17}$$

taking $\boldsymbol{a}_{(k-1)/2+1}$ to the zero-COM hyper-plane, while $\boldsymbol{x}$ and every other augmented variable remain non-zero-CoM. Before the second coupling transform $\mathcal{F}_M^{(2)}$, we apply the $\mathrm{ShiftCoM}$ maps in reverse, taking $\boldsymbol{x}$ back to the zero-COM hyper-plane.

The coupling transforms $\mathcal{F}_M^{(1)}$ and $\mathcal{F}_M^{(2)}$ work as previously, except now they transform $(k+1)/2$ sets of variables (holding the other set constant), where for each of the $k$ variables, each of the $n$ nodes get their own local reference frame and inner flow transform $\tau_\theta$.

## B.5  Invariant base and target distributions

We now show the $\mathrm{SE}(3) \times S_n$ invariance of our extended base distribution choice, which trivial extends to our choice of augmented target distribution. Consider the choice $\pi(\boldsymbol{a}|\boldsymbol{x}) = \mathcal{N}(\boldsymbol{a}; \boldsymbol{x}, \eta^2 I)$ for $\eta^2$ a scalar hyperparameter. With this choice, the constraint of interest is satisfied since the Isotropic Gaussian is rationally invariant and centered at $\boldsymbol{x}$. To see this, let $t \in \mathrm{SE}(3)$ be decomposed

into two components $t = (R, u)$ where $R \in \mathrm{SO}(3)$ is a $3 \times 3$ rotation matrix and $u \in \mathbb{R}^3$ represents a translation,

$$
\begin{aligned}
\mathcal{N}(R\boldsymbol{a} + u;\ R\boldsymbol{x} + u,\ \eta^2 I) &= \frac{1}{Z} \exp \frac{-1}{2} (R\boldsymbol{a} + u - R\boldsymbol{x} + u)^\top (\eta^{-2} I)(R\boldsymbol{a} + u - R\boldsymbol{x} + u) \quad (18) \\
&= \frac{1}{Z} \exp \frac{-1}{2} (\boldsymbol{a} - \boldsymbol{x})^\top R^\top (\eta^{-2} I) R (\boldsymbol{a} - \boldsymbol{x}) \\
&= \frac{1}{Z} \exp \frac{-1}{2} (\boldsymbol{a} - \boldsymbol{x})^\top R^\top R (\eta^{-2} I)(\boldsymbol{a} - \boldsymbol{x}) \\
&= \mathcal{N}(\boldsymbol{a};\ \boldsymbol{x},\ \eta^2 I) \quad (19)
\end{aligned}
$$

An analogous argument holds for permutations, since any $\sigma \in S_n$ can be represented as a unitary $n \times n$ matrix which is simultaneously applied to $\boldsymbol{a}$ and $\boldsymbol{x}$.

### B.6 Maximum likelihood objective

We train our models from samples by maximising the joint log-density $\mathbb{E}_{\boldsymbol{x} \sim p(\boldsymbol{x}), \boldsymbol{a} \sim \pi(\boldsymbol{a}|\boldsymbol{x})}[\log q(\boldsymbol{x}, \boldsymbol{a})]$. This can be shown to be a lower bound on the expected marginal log-density over observations, which is the quantity targeted by models without augmented variables as

$$
\mathrm{E}_{p(\boldsymbol{x})}[\log q(\boldsymbol{x})] = \mathrm{E}_{p(\boldsymbol{x})} \left[ \log \mathrm{E}_{\pi(\boldsymbol{a}|\boldsymbol{x})} \left[ \frac{q(\boldsymbol{x}, \boldsymbol{a})}{\pi(\boldsymbol{a}|\boldsymbol{x})} \right] \right] \geq \mathrm{E}_{p(\boldsymbol{x})\pi(\boldsymbol{a}|\boldsymbol{x})} \left[ \log q(\boldsymbol{x}, \boldsymbol{a}) - \log \pi(\boldsymbol{a}|\boldsymbol{x}) \right] \quad (20)
$$

where we drop the latter term from our training objective since it is constant w.r.t. flow parameters. The bound becomes tight when $q(\boldsymbol{a}|\boldsymbol{x}) = \pi(\boldsymbol{a}|\boldsymbol{x})$.

### B.7 Reverse KL objective

We now show the reverse KL objective involving the joint distribution of observations and augmented variables presented in the main text upper bounds the reverse KL over observations. Using the chain rule of the KL divergence we have

$$
\begin{aligned}
D_{\mathrm{KL}}\left(q(\boldsymbol{x}, \boldsymbol{a}) \,\|\, p(\boldsymbol{x})\pi(\boldsymbol{a}|\boldsymbol{x})\right) &= D_{\mathrm{KL}}\left(q(\boldsymbol{x}) \,\|\, p(\boldsymbol{x})\right) + \mathbb{E}_{q(\boldsymbol{x})} D_{\mathrm{KL}}\left(q(\boldsymbol{a}|\boldsymbol{x}) \,\|\, \pi(\boldsymbol{a}|\boldsymbol{x})\right) \\
&\geq D_{\mathrm{KL}}\left(q(\boldsymbol{x}) \,\|\, p(\boldsymbol{x})\right).
\end{aligned}
$$

The looseness in the bound is $\mathbb{E}_{q(\boldsymbol{x})} D_{\mathrm{KL}}\left(q(\boldsymbol{a}|\boldsymbol{x}) \,\|\, \pi(\boldsymbol{a}|\boldsymbol{x})\right)$ which again vanishes when $q(\boldsymbol{a}|\boldsymbol{x}) = \pi(\boldsymbol{a}|\boldsymbol{x})$.

### B.8 Alpha divergence objective

We now show that our joint objective represents an analogous upper bound for the $\alpha$ divergence when $\alpha = 2$, i.e., the objective targeted by FAB [Midgley et al., 2023]. Let $U(\boldsymbol{x})$ be the unnormalized energy function of interest evaluated at state $\boldsymbol{x}$, with $p(\boldsymbol{x}) \propto \exp(-U(\boldsymbol{x}))$. We have

$$
\begin{aligned}
D_2\left(q(\boldsymbol{x}) \,\|\, p(\boldsymbol{x})\right) &= \int \frac{p(\boldsymbol{x})^2}{q(\boldsymbol{x})} \, \mathrm{d}\boldsymbol{x} \\
&\leq \int \frac{p(\boldsymbol{x})^2}{q(\boldsymbol{x})} \left( \int \frac{\pi(\boldsymbol{a}|\boldsymbol{x})^2}{q(\boldsymbol{a}|\boldsymbol{x})} \, \mathrm{d}\boldsymbol{a} \right) \mathrm{d}\boldsymbol{x} \\
&= \int \frac{p(\boldsymbol{x})^2 \pi(\boldsymbol{a}|\boldsymbol{x})^2}{q(\boldsymbol{x}) q(\boldsymbol{a}|\boldsymbol{x})} \, \mathrm{d}\boldsymbol{a} \, \mathrm{d}\boldsymbol{x} \\
&= D_2\left(q(\boldsymbol{x}, \boldsymbol{a}) \,\|\, p(\boldsymbol{x})\pi(\boldsymbol{a}|\boldsymbol{x})\right)
\end{aligned}
$$

which also becomes tight when $q(\boldsymbol{a}|\boldsymbol{x}) = \pi(\boldsymbol{a}|\boldsymbol{x})$. The inequality is true because $\int \frac{\pi(\boldsymbol{a}|\boldsymbol{x})^2}{q(\boldsymbol{a}|\boldsymbol{x})} \, \mathrm{d}\boldsymbol{a} > 0$. To see this, take any densities $\pi$ and $q$, and apply Jensen's inequality as

$$
\int \frac{\pi(a)^2}{q(a)} \, \mathrm{d}a = \mathrm{E}_{\pi(a)} \frac{\pi(a)}{q(a)} = \mathrm{E}_{\pi(a)} \left( \frac{q(a)}{\pi(a)} \right)^{-1} \geq \left( \mathrm{E}_{\pi(a)} \frac{q(a)}{\pi(a)} \right)^{-1} = \int q(a) \mathrm{d}a = 1.
$$

## B.9 Importance weighted estimator

We define the following estimator $w := \frac{q(\boldsymbol{x}, \boldsymbol{a})}{\pi(\boldsymbol{a}|\boldsymbol{x})}$ with $\boldsymbol{a} \sim \pi(\cdot|x)$. Assuming that whenever $\pi(\boldsymbol{a}|\boldsymbol{x}) = 0$ then $q(\boldsymbol{x}, \boldsymbol{a}) = 0$, we have that this estimator is trivially unbiased:

$$
\begin{aligned}
\mathbb{E}[w] &= \int \frac{q(\boldsymbol{x}, \boldsymbol{a})}{\pi(\boldsymbol{a}|\boldsymbol{x})} \pi(\boldsymbol{a}|\boldsymbol{x}) d\mu(\boldsymbol{a}) \\
&= \int q(\boldsymbol{x}, \boldsymbol{a}) d\mu(\boldsymbol{a}) \\
&= q(\boldsymbol{x})
\end{aligned}
$$

and that is variance is given by

$$
\begin{aligned}
\mathbb{V}[w] &= \mathbb{E}[(w - q(\boldsymbol{x}))^2] \\
&= \mathbb{E}[w^2] - \mathbb{E}[2wq(\boldsymbol{x})] + \mathbb{E}[q(\boldsymbol{x})^2] \\
&= \int \frac{q(\boldsymbol{x}, \boldsymbol{a})^2}{\pi(\boldsymbol{a}|\boldsymbol{x})^2} \pi(\boldsymbol{a}|\boldsymbol{x}) d\mu(\boldsymbol{a}) - 2q(\boldsymbol{x})\mathbb{E}[w] + q(\boldsymbol{x})^2 \\
&= \int \frac{q(\boldsymbol{x})q(\boldsymbol{a}|\boldsymbol{x})}{\pi(\boldsymbol{a}|\boldsymbol{x})} q(\boldsymbol{x}, \boldsymbol{a}) d\mu(\boldsymbol{a}) - q(\boldsymbol{x})^2
\end{aligned}
$$

thus if $q(\boldsymbol{a}|\boldsymbol{x}) = \pi(\boldsymbol{a}|\boldsymbol{x})$, $\mathbb{V}[w] = 0$.

The log marginal likelihood may therefore be estimated with

$$
\log q(\boldsymbol{x}) \approx \log \frac{1}{M} \sum_{m=1}^{M} \frac{q(\boldsymbol{x}, \boldsymbol{a}_m)}{\pi(\boldsymbol{a}_m|\boldsymbol{x})} \quad \text{and} \quad \boldsymbol{a}_m \sim \pi(\cdot|\boldsymbol{x}), \tag{21}
$$

which becomes accurate as M becomes large.

## B.10 Effective sample size

For evaluation we use both the reverse and forward effective sample size (ESS), where the former relies on samples from the model, and the latter from samples from the target. For the augmented flow models we report the joint ESS, which lower bounds the marginal ESS. Below, we provide the formulas for the reverse and forward ESS. For simplicity, we define these in terms of the marginal ESS. The joint ESS estimators may be obtained using the same formulas, replacing $\boldsymbol{x}$ with $(\boldsymbol{x}, \boldsymbol{a})$.

The reverse effective sample size is given by

$$
n_{\text{e,rv}} = \frac{(\sum_{i=1}^{n} w(\boldsymbol{x}_i))^2}{\sum_{i=1}^{n} w(\boldsymbol{x}_i)^2} \quad \text{and} \quad \boldsymbol{x}_i \sim q(\boldsymbol{x}_i), \tag{22}
$$

where $w(\boldsymbol{x}_i) = \tilde{p}(\boldsymbol{x}_i)/q(\boldsymbol{x}_i)$ and $\tilde{p}$ is the unnormalized version of $p$ [Owen, 2013].

The forward ESS may be estimated with

$$
n_{\text{e,fwd}} = \frac{n^2}{Z^{-1} \sum_{i=1}^{n} w(\boldsymbol{x}_i)} \quad \text{and} \quad \boldsymbol{x}_i \sim p(\boldsymbol{x}_i), \tag{23}
$$

where the normalizing constant $Z = \tilde{p}/p$ may be estimated using

$$
Z^{-1} = E_p \left[ \frac{q(\boldsymbol{x})}{\tilde{p}(\boldsymbol{x})} \right] \tag{24}
$$

$$
\approx \frac{1}{n} \sum_{i=1}^{n} \frac{q(\boldsymbol{x}_i)}{\tilde{p}(\boldsymbol{x}_i)} \quad \text{and} \quad \boldsymbol{x}_i \sim p(\boldsymbol{x}_i). \tag{25}
$$

## B.11 Conversion of internal coordinate flow density to density on ZeroCoM Cartesian space with global rotation represented

The density of the flow on internal coordinates is not directly comparable to the flows on Cartesian coordinates, as the variables live in a different space (bond distances, angles and torsions instead

of Cartesian coordinates). The internal coordinates, and density may be converted into Cartesian coordinates via passing them through the appropriate bijector and computing the log determinant of the bijector. This is commonly done in existing literature as the target energy typically takes Cartesian coordinates as input [Midgley et al., 2023]. However, this representation is invariant to global rotations of the molecule, making it lower dimensional than the space with which the flows which introduced in this paper operate. Furthermore, this representation has the first atom centered on the origin, where the flows in this paper instead have zero-CoM. To compare the density of the flow on internal coordinates with the flows that operate directly on Cartesian coordinates we have to account for both of these aforementioned factors. We describe how to do this below.

**Conversion to density that includes degrees of freedom representing global rotation**   The rotation-invariant representation of internal coordinates in Cartesian space places the first atom at the origin, the second atom a distance of $b_1$ along the X-axis (i.e. at position $\{b_1, 0, 0\}$), and the third atom on the XY plane, at a distance of $b_2$ from the origin at an angle of $a2$ with respect to the x-axis (i.e. at position $\{\cos(a_2) \cdot b_2, \sin(a_2) \cdot b_2, 0\}$). We define this representation of Cartesian coordinates as $z \in Z$, where we drop all coordinates of the first atom (atom 0), the $y$ and $z$ coordinates of atom 1, and the $z$ coordinate of atom 2, all of which are zero, so that $Z$ contains as many elements as there are degrees of freedom.

To convert this into a space that has degrees of freedom representing global rotations, which we refer to as $w \in W$, we simply sample a rotation uniformly over $SO3$ and apply this to $w$.

The rotation has three degrees of freedom, which we represent as follows; First, we apply rotation about the X-axis of $o_1$ sampled uniformly over $[-\pi, \pi]$. Secondly, we rotate about the Z-Axis by sampling $o_2$ uniformly between $[-1, 1]$ and rotating by $\sin^{-1}(o_2)$. Finally, we rotate about the Y-axis by $o_3$ sampled uniformly between $[-\pi, \pi]$. Together this ensures that the resultant orientation of the molecule in Cartesian space has a uniform density over $SO3$. We denote the concatenation of these random variables that parameterize the rotation as $\boldsymbol{o} \in \mathcal{O}$ where $\mathcal{O} = \{[-\pi, \pi], [-1, 1], [-\pi, \pi]\}$.

The density on $W$ is then given by

$$q(\boldsymbol{w}) = q(\boldsymbol{z})q(\boldsymbol{o}) \left| \det \frac{\partial g(\boldsymbol{z}, \boldsymbol{o})}{\partial \{\boldsymbol{z}, \boldsymbol{o}\}} \right|^{-1}. \tag{26}$$

where $g : \mathcal{Z} \times \mathcal{O} \to \mathcal{W}$ is a bijector which applies the rotation parameterized by $o$ to $z$.

It can be shown that the log jacobian determinant of $g$ is $2\log(b_1) + \log(b_2) + \log(\sin(a_2))$. Furthermore, $\log q(\boldsymbol{o})$ is a constant value of $-\log 8\pi$.

**Conversion density on Zero-CoM space**   After applying the above transform we have $\boldsymbol{w}$ that includes rotations, but is centered on atom 0. Thus, the last step needed is to convert into the Zero-CoM space. As previouly, to ensure that our representations have the same number of elements as there are degrees of freedom, we drop the first atom's Cartesian coordinates, as these contain redundant information. We note that for the representation $\boldsymbol{w} \in \mathcal{W}$ the first atom's coordinates are at the origin, and for the Zero-CoM space $\boldsymbol{x} \in \tilde{\mathcal{X}}$ they are at $-\sum_{i=1}^{n-1} x_i$. The bijector $h : \mathcal{W} \to \tilde{\mathcal{X}}$ is given by

$$h(\boldsymbol{w}) = \boldsymbol{w} - \frac{1}{N} \sum_{i=1}^{n-1} \boldsymbol{w}_i \tag{27}$$

where $N$ is the number of nodes. The log jacobian determinant of $h$ is simply equal to $-3\log n$.

**Overall conversion**   Thus we can convert the density in $\mathcal{Z}$ into an equivalent density in $\tilde{\mathcal{X}}$ using,

$$\log q(\boldsymbol{x}) = \log q(\boldsymbol{z}) + \log q(\boldsymbol{o}) + \log \left| \det \frac{\partial g(\boldsymbol{z}, \boldsymbol{o})}{\partial \{\boldsymbol{z}, \boldsymbol{o}\}} \right|^{-1} + \log \left| \det \frac{\partial h(\boldsymbol{w})}{\partial \boldsymbol{w}} \right|^{-1}$$
$$= \log q(\boldsymbol{z}) - \log 8\pi^2 - 2\log(b_1) - \log(b_2) - \log(\sin(a_2)) + 3\log n. \tag{28}$$

## C   Experiments: Details and Further Results

We provide the details of the hyper-parameters and compute used for each experiment. Additionally, we provide the code https://github.com/lollcat/se3-augmented-coupling-flows

for replicating our experiments. It also contains the configuration files with the exact hyperparameter for each model and training run.

## C.1 Custom Optimizer

For all experiments we use a custom defined optimizer, which improves training stability by skipping updates with very large gradient norms, and clipping updates with a moderately large gradient norms. Our optimizer keeps track of a window the the last 100 gradient norms, and then for a given step: (1) If the gradient norm is more than 20 times greater than the median gradient norm within the window, then skip the gradient step without updating the parameters. (2) Otherwise, if the gradient norm is more than five times the median within the window then clip the gradient norm to five times the median. (3) Use the (possibly clipped) gradients to perform a parameter update using the adam optimizer [Kingma and Ba, 2014]. Lastly, we also include an optional setting which allows for masking of gradient contribution of samples within each batch with very large negative log probability during training, which further improves training stability.

## C.2 EGNN implementation

We use our own custom implementation of the EGNN proposed by Satorras et al. [2021b] for our experiments. We found that applying a softmax to the invariant features output from the EGNN helped improve stability. Additionally, we extend Satorras et al. [2021b]'s EGNN to allow for multiple channels of positions/vectors at each node, with interaction across channels. This is needed as the local reference frame $r$ typically is parameterized by multiple equivariant vectors, and for the case where the multiple sets of augmented variables are used such that there are multiple sets of positions input to the EGNN. We refer to the code for the details of our EGNN implementation.

## C.3 Training with samples

### C.3.1 DW4, LJ13 and QM9 Positional

**Experimental details** Below we provide a description of the hyper-parameters used within each experiment. Additionally Tab. 4 provides the run-times for each experiment.

**Datasets**: For DW4 and LJ13 we use a training and test set of 1000 samples following Satorras et al. [2021a]. We note that this data is biased with a clear difference between the distribution of train and test data (see Fig. 5. For QM9-positional we use a training, validation and test set of 13831, 2501 and 1,813 samples respectively also following Satorras et al. [2021a].

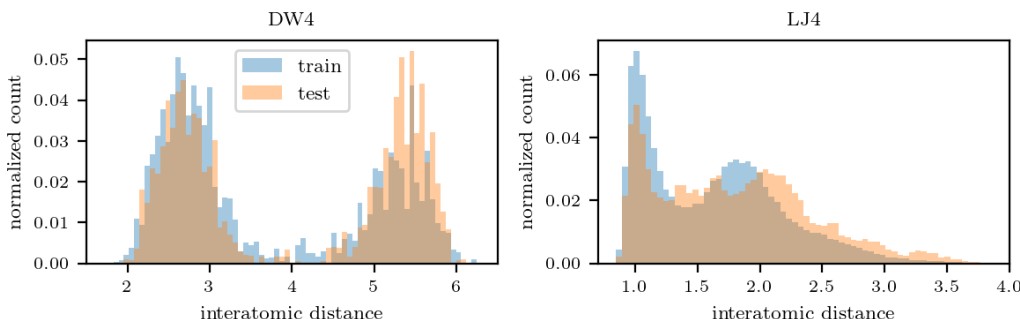

Figure 5: Inter-atomic distances for samples from the train and test data on DW4 and LJ13. Both datasets are biased with visible differences between the train and test sets.

**Flow experiment details** All flow models using 12 blocks (see Alg. 1 for the definition of a block). For the noise scale of the augmented variables sampled from $\mathcal{N}(\boldsymbol{a}; \boldsymbol{x}, \eta^2 I)$, we use $\eta = 0.1$ for both the base of the flow, and augmented target distribution. For the CARTESIAN-PROJ E-ACF and NON-E-ACF we use RealNVP [Dinh et al., 2017] for the flow transform, while for CARTESIAN-PROJ and VECTOR-PROJ we use a rational quadratic spline transform [Durkan et al., 2019]. For the spline we use 8 bins for all flows. We use 1 set of augmented variables for DW4 and LJ13, and 3 sets

of augmented variables for QM9-positional. For all equivariant GNN's we implemented the GNN proposed by Satorras et al. [2021b] with 3 blocks. All MLP's have 2 layers, with a width of 64 units. For the NON-E-ACF model we use a transformer with 3 layers of self-attention with 8 heads. We use a cosine learning rate schedule, that is initialised to a value of 0.00002, peaks at 0.0002 after 30 epochs, and then decays back to a final learning rate of 0.00002 for DW4 and LJ13. For QM9 the learning rate is initialised to a value of 0.00006, peaks at 0.0002 after 30 epochs, and then decays back to a final learning rate of 0.00006. With the above learning rate schedule for QM9 the CARTESIAN-PROJ E-ACF was unstable during training which resulted in crashed runs (NaN loss). Hence, it was instead trained with a constant learning rate of 0.00006 at which training was stable. We use a batch size of 32 for all experiments, and 100, 400 and 800 epochs of the datasets for DW4, LJ13 and QM9-positional respectively. We train the NON-E-ACF for 4 times as many epochs for all problems, which results in roughly the same run time as the SPHERICAL-PROJ E-ACF. For the CARTESIAN-PROJ and SPHERICAL-PROJ E-ACF, we use a coefficient of 10. for the auxiliary loss (Eq. (11)), for weighting it relative to the maximum likelihood loss. Marginal log likelihood in Tab. 1 is calculated using Eq. (21) with 20 samples from the augmented variables $a$ per test-set sample of $x$.

**Continuous normalizing flow / diffusion experiment details** For the E-CNF ML we report the results from Satorras et al. [2021a]. For the E-CNF FM we implement flow matching, with the model's vector field parameterized by the E(n) GNN of Satorras et al. [2021b]. For DW4 and LJ13 we use an EGNN with 3 message passing layers, with a 3-hidden layer MLP with 128 units per layer. For QM9-positional we use an EGNN with 5 message passing layers, with an 4-hidden layer MLP with 256 units per layer. For all problems we use Adam optimizer with a cosine learning rate decay from 1e-4 to 0. Additionally, for QM9, we use EMA with $\beta = 0.999$ to compute the final training weights which the model is evaluated with. We train for 200, 400, and 1600 epochs on DW4, LJ13 and QM9-positional respectively. For sampling and evaluating log-densities we use the `dopri5` ODE solver [Dormand and Prince, 1980] with an `rtol` and `atol` of 1e-5. For DW4 and LJ13 we evaluate densities using the exact divergence of the vector field, while for QM9 we approximate the divergence using the Hutchinson trace estimator [Grathwohl et al., 2018] as the exact divergence is computationally expensive.

Our E-CNF-DIFF implementation roughly follows Hoogeboom et al. [2022], but we use the noise schedule as Ho et al. [2020]. We use the same GNN architecture as for our flows, that is the E(n) GNN of Satorras et al. [2021b], to predict the diffusion noise. The network is composed of 3 message passing layers, each of which uses a 3 hidden layer MLP with 256 units per layer. GNN expands its inputs to 256 scalars and 196 vectors per graph node and returns its predictions as a linear combination of the latter 196 vectors. This model has around 90M learnable parameters. We use the AdamW optimiser with a weight decay of $10^{-4}$ and run it for 100, 200, and 5000 epochs for the DW4, LJ13 and QM9 dataset, respectively. The initial learning rate is set to $10^{-5}$. It is linearly increased to $5 \cdot 10^{-4}$ for the first 15% of training before being decreased to $10^{-6}$ with a cosine schedule. Following, Kingma et al. [2021], we sample noise timesteps during such that these are uniformly spaced within each batch, reducing gradient variance. We set the smallest accessible time value to $10^{-3}$. For sampling and evaluating log-densities we use the `dopri5` ODE solver with a maximum number of evaluations of 1000.

Table 4: Run time for training by maximum likelihood on the DW4, LJ13 and QM9-positional datasets, in hours. All runs used an GeForce RTX 2080Ti GPU . Run times include evaluation that was performed intermittently throughout training. The results are averaged over 3 seeded runs, with the standard error reported as uncertainty.

|  | DW4 | LJ13 | QM9 positional |
|---|---|---|---|
| E-CNF ML | $N/A \pm N/A$ | $N/A \pm N/A$ | $336 \pm N/A$ |
| E-CNF-DIFF | $N/A \pm N/A$ | $N/A \pm N/A$ | $3.80 \pm 0.00$ |
| NON-E-ACF | $0.12 \pm 0.00$ | $0.54 \pm 0.00$ | $20.68 \pm 0.09$ |
| VECTOR-PROJ E-ACF | $0.16 \pm 0.00$ | $0.61 \pm 0.00$ | $19.98 \pm 0.04$ |
| CARTESIAN-PROJ E-ACF | $0.12 \pm 0.00$ | $0.50 \pm 0.00$ | $17.13 \pm 0.33$ |
| SPHERICAL-PROJ E-ACF | $0.15 \pm 0.00$ | $0.59 \pm 0.00$ | $20.61 \pm 0.17$ |

**Further results** Tab. 5 provides the effective sample size of the flows trained by maximum likelihood, with respect to the joint target distribution $p(x)\pi(a|x)$.

Table 5: Joint effective sample size (%) evaluated using 10000 samples. The results are averaged over 3 seeded runs, with the standard error reported as uncertainty.

| | DW4 | | LJ13 | |
| | Rev ESS (%) | Fwd ESS (%) | Rev ESS (%) | Fwd ESS (%) |
| --- | --- | --- | --- | --- |
| E-CNF FM | $5.02 \pm 2.22$ | $0.01 \pm 0.00$ | $5.37 \pm 0.35$ | $0.30 \pm 0.18$ |
| NON-E-ACF | $0.68 \pm 0.18$ | $0.06 \pm 0.02$ | $0.00 \pm 0.00$ | $0.00 \pm 0.00$ |
| VECTOR-PROJ E-ACF | $1.87 \pm 0.68$ | $0.16 \pm 0.01$ | $1.80 \pm 0.17$ | $0.06 \pm 0.03$ |
| CARTESIAN-PROJ E-ACF | $3.08 \pm 0.05$ | $0.09 \pm 0.03$ | $0.89 \pm 0.37$ | $0.00 \pm 0.00$ |
| SPHERICAL-PROJ E-ACF | $3.06 \pm 0.60$ | $0.17 \pm 0.05$ | $1.50 \pm 0.37$ | $0.02 \pm 0.00$ |

**Varying the number of training samples for DW4 and LJ13**   Tab. 6 shows the effect of varying the training dataset size on the performance of the flows. We tune the number of epochs via the validation set; we use 1000, 100, 50 and 35 epochs for DW4 and 1500, 1500, 400 and 100 LJ13 for each dataset size respectively. Additionally we train NON-E-ACF (with data augmentation) for 4 times as many epochs for each experiment, as it is computationally cheaper per epoch and takes a larger number of epochs before it overfits. We see that the performance gap between the non-equivariant and equivariant flow models is larger in the lower data regime.

Table 6: Negative log likelihood results on test set for flows trained by maximum likelihood for the DW4 and LJ13 datasets varying the number of training samples. E-CNF results are from Satorras et al. [2021a]. Best results are emphasized in **bold**.

| | DW4 | | | | LJ13 | | | |
| # samples | $10^2$ | $10^3$ | $10^4$ | $10^5$ | 10 | $10^2$ | $10^3$ | $10^4$ |
| --- | --- | --- | --- | --- | --- | --- | --- | --- |
| E-CNF | **8.31** | **8.15** | **7.69** | 7.48 | **33.12** | 30.99 | 30.56 | 30.41 |
| NON-E-ACF | 10.24 | 10.07 | 8.23 | 7.39 | 40.21 | 34.83 | 33.32 | 31.94 |
| VECTOR-PROJ E-ACF | 8.65 | 8.69 | 7.95 | 7.47 | 33.64 | **30.61** | **30.19** | **30.09** |
| CARTESIAN-PROJ E-ACF | 8.79 | 8.82 | 7.93 | 7.52 | 36.66 | 31.72 | 30.89 | 30.48 |
| SPHERICAL-PROJ E-ACF | 8.66 | 8.61 | 7.97 | **7.38** | 34.79 | 31.26 | 30.33 | 30.16 |

### C.3.2   Alanine dipeptide

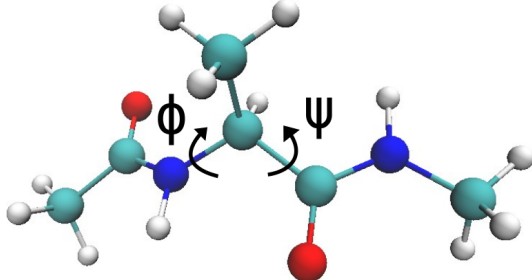

Figure 6: Visualization of the 22-atom molecule alanine dipeptide with its two dihedral angles $\phi$ and $\psi$.

The molecule alanine dipeptide and its dihedral angles $\phi$ and $\psi$, which are used in the Ramachandran plots in Fig. 3, are depicted in Fig. 6. We model it in an implicit solvent at a temperature of $T = 800\,\mathrm{K}$. The data used for maximum likelihood training and testing was generated with a replica exchange MD simulation [Mori and Okamoto, 2010] with 21 systems having temperatures from $300\,\mathrm{K}$ until $1300\,\mathrm{K}$ with $50\,\mathrm{K}$ temperature difference between each of them.

Each ACF has 20 layers with 3 core transformations per layer for the VECTOR-PROJ flow and 1 for the CARTESIAN-PROJ and the SPHERICAL-PROJ flow. The flow trained on internal coordinates has the same architecture as the models in Midgley et al. [2023], i.e. it is a coupling neural spline flow with 12 layers. The models were trained for 50 epochs.

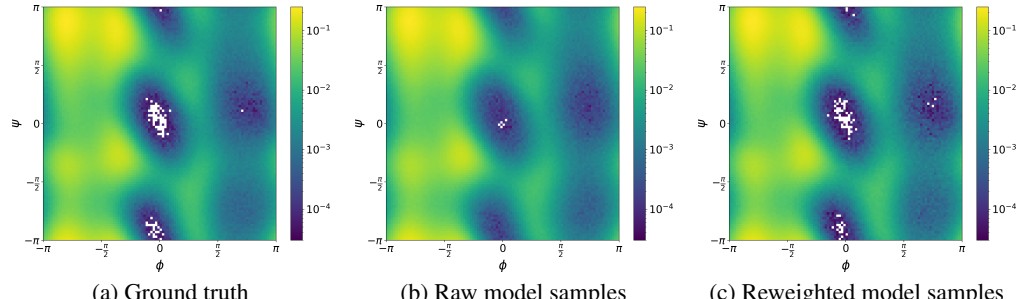

|                | (a) Ground truth | (b) Raw model samples | (c) Reweighted model samples |
|----------------|------------------|------------------------|------------------------------|

Figure 7: Ramachandran plot of a SPHERICAL-PROJ E-ACF model trained on alanine dipeptide at 800 K. The samples in Fig. 7c are reweighted with the importance weights. Thereby, the KLD drops from $1.88 \cdot 10^{-3}$ to $1.39 \cdot 10^{-3}$.

The E-CNF and the E-CNF-DIFF model have E(n) GNN with 4 message passing layers, each using a 3-layered MLP with 128 hidden units each, as a vector field and score function, respectively. Both were trained for 20 epochs.

Since the version of the VECTOR-PROJ flow as well as the E-CNF and E-CNF-DIFF models are parity equivariant, the model learns to generate both chiral forms of alanine dipeptide [Midgley et al., 2023]. For evaluating the model, we filtered the samples from the model to only include those that correspond to the L-form, which is the one almost exclusively found in nature.

To plot and compute the KLD of the Ramachandran plots, we drew $10^7$ samples from each model. Since computing the likelihood of the E-CNF and the E-CNF-DIFF model is so expensive, we evaluated these models only on 10% of the test set, i.e. $10^6$ datapoints.

**Limiting the number of samples** To test how well E-ACFs generalizes thanks to their equivariance, we trained our three variants of E-ACFs and an NON-E-ACFon subsets of the training dataset with various sizes. The results are given in Table 7. For all training dataset sizes the equivariant architectures clearly outperform the non-equivariant counterpart. Notably, the NON-E-ACF trained on the full dataset performs worse than the VECTOR-PROJ E-ACF trained on 10% of the data and the CARTESIAN-PROJ and SPHERICAL-PROJ E-ACF trained on just 1% of the data.

Table 7: NLL of the models trained on subsets of the training dataset with different sizes. The best performing model, i.e. the one with the lowest NLL, is marked in **bold** for each training dataset size.

| Training dataset size | $10^2$ | $10^3$ | $10^4$ | $10^5$ | $10^6$ |
|-----------------------|--------|--------|--------|--------|--------|
| NON-E-ACF             | $-28.8$ | $-76.8$ | $-120.4$ | $-164.3$ | $-177.8$ |
| VECTOR-PROJ E-ACF     | $-31.0$ | $-111.4$ | $-169.5$ | $-183.6$ | $-188.5$ |
| CARTESIAN-PROJ E-ACF  | $\mathbf{-65.1}$ | $\mathbf{-155.7}$ | $\mathbf{-185.9}$ | $-188.1$ | $\mathbf{-188.6}$ |
| SPHERICAL-PROJ E-ACF  | $-53.1$ | $-119.3$ | $-183.6$ | $\mathbf{-188.3}$ | $\mathbf{-188.6}$ |

**Reweighting** Given the high effective sample sizes we obtained for our models (see Tab. 2), we also attempted to reweight the samples. As already observed by Midgley et al. [2023], for a large number of samples, i.e. $10^7$, there were a few high outlier importance weights, which spoil the importance weight distribution. Hence, we clipped the highest $0.2\%$ of the importance weights as in [Midgley et al., 2023].

The results for our SPHERICAL-PROJ E-ACF model are shown in Fig. 7. The reweighted Ramachandran plot is significantly closer to the ground truth in low density regions. The KLD is reduced from $1.88 \cdot 10^{-3}$ to $1.39 \cdot 10^{-3}$.

**Alanine dipeptide at $T = 300$K** Although our experiments were done on alanine dipeptide at $T = 800$K, we also tried $T = 300$K. We trained our SPHERICAL-PROJ E-ACF model on the same

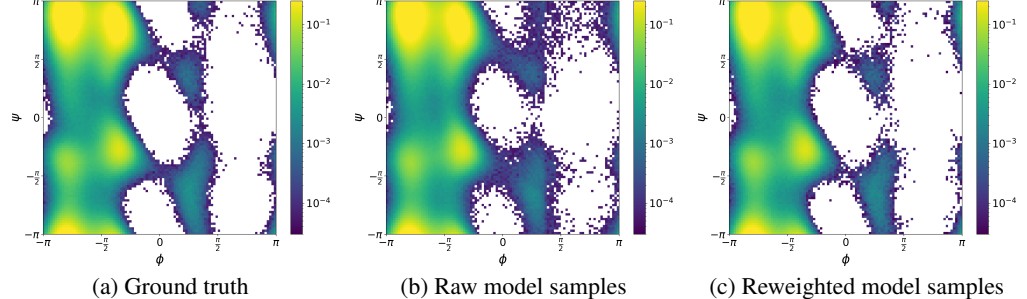

| (a) Ground truth | (b) Raw model samples | (c) Reweighted model samples |

Figure 8: Ramachandran plot of a SPHERICAL-PROJ E-ACF model trained on alanine dipeptide at $300\,\mathrm{K}$. The raw and reweighted results are shown. Through reweighting, the KLD drops from $2.79 \cdot 10^{-3}$ to $1.97 \cdot 10^{-3}$.

dataset used in [Midgley et al., 2023]. As shown in Fig. 8, we got good performance. Even without reweighting our model has a KLD of $2.79 \cdot 10^{-3}$. When doing reweighting, it decreases to $1.97 \cdot 10^{-3}$. For comparison, the flows trained on internal coordinates presented in [Midgley et al., 2023] had a KLD of $(7.57 \pm 3.80) \cdot 10^{-3}$, and reweighting worsened the results given the low ESS. However, these models had about 6 times fewer parameters than our model.

### C.4  Energy-based training

For both the DW4 and LJ13 experiments we use an identical flow architecture and optimizer setup to the flows trained by maximum likelihood with samples App. C.3.1. We use a batch size of 128 and 1024 for DW4 and LJ13 respectively. We train for 20000 and 14000 epochs for DW4 and LJ13 for the E-ACF. For the NON-E-ACF we train for 60000 and 56000 epochs for DW4 and LJ13 respectively, which corresponds to roughly the same training time as the SPHERICAL-PROJ E-ACF because the NON-E-ACF is faster per iteration. Reverse ESS is estimated with 10,000 samples from the flow. Forward ESS is estimated with the test sets.

**Hyper-parameters for FAB**: We set the minimum replay buffer size to 64 batches and maximum size to 128 batches. For the importance weight adjustment for re-weighting the buffer samples to account for the latest flow parameters, we clip to a maximum of 10. For AIS we use HMC with 5 leapfrog steps, and 2 and 8 intermediate distributions for DW4 and LJ13 respectively. We tune the step size of HMC to target an acceptance probability of 65%. We perform 8 buffer sampling and parameter update steps per AIS forward pass.

Table 8: Run time for energy-based training with FAB on the DW4 and LJ13 problems. DW4 was run using a GeForce RTX 2080Ti GPU and LJ13 with a V-4 TPU. Run times include evaluation that was performed intermittently throughout training. The results are averaged over 3 seeded runs, with the standard error reported as uncertainty.

|  | DW4 | LJ13 |
| --- | --- | --- |
| NON-E-ACF | $8.9 \pm 0.0$ | $38.1 \pm 0.0$ |
| VECTOR-PROJ E-ACF | $7.5 \pm 0.0$ | $45.0 \pm 0.1$ |
| CARTESIAN-PROJ E-ACF | $6.3 \pm 0.0$ | $42.7 \pm 0.0$ |
| SPHERICAL-PROJ E-ACF | $7.3 \pm 0.0$ | $44.7 \pm 0.0$ |

