# OpenReview forum: "SE(3) Equivariant Augmented Coupling Flows"
_NeurIPS.cc/2023/Conference — NeurIPS 2023 spotlight_

### Official Review · Reviewer_Xvtr · 2023-06-22

**Soundness:** 4 excellent
**Presentation:** 4 excellent
**Contribution:** 3 good
**Rating:** 6
**Confidence:** 3

**Summary:**

The manuscript proposes a normalizing flow architecture that is i.) equivariant wrt to the euclidian group SE(3) and permutation group S_n and ii.) is based on coupling blocks. Existing work on this specific equivariance has relied on continuous normalizing flows which rely on a Neural ODE instead of a coupling architecture to construct a bijective map.

Coupling-based architectures have the advantage that they typically faster - both in training and sampling phase.

The proposed architecture combines various previously explored ideas in an elegant manner. Specifically, it relies on augmented flows to realise the equivariance, similar to Klein et al and first proposed in by Huang et al., and achieves SE(3) symmetry by relying on zero centre of mass subspaces.

The experimental part gives a detailed comparison to existing baselines and demonstrates the advantage of the proposed architecture over CNF-based flows as well as the proposal by Klein et al.

Overall, I found this paper an insightful and solid contribution as well as a nice read. The proposed architecture is definitely worth exploring in my own research. I would encourage the authors to publish their code to maximize the impact of their paper.


**Strengths:**

- Very clearly written manuscript
- A thorough and extensive experimental analysis
- Proposed architecture is a elegant combination of well-known mechanisms achieving full equivariance
- Both likelihood and energy-based training are considered

**Weaknesses:**

- The ms mainly combines previously proposed ideas (albeit in an innovative and elegant fashion). In particular, its proposal is relatively similar to the one of Klein et al.



**Questions:**

- How is the KLD normalizer in the tables estimated?
- Did you investigate if any mode-collapse is present for the reverse KL training? In particular, did you estimate the forward ESS, i.e. 1/E_p w where p is the target/data density and w the importance weight?
- Cartesian projection seems to lead to numerical instabilities. Do you have an understanding of why they arise?

**Limitations:**

Yes, I think the limitations are sufficiently summarised although I would have appreciated a dedicated section.

---

> ### Author Rebuttal · Authors · 2023-08-08
>
> Thank you for your review - we appreciated that you find our work relevant to your research! Below we have responded to your feedback.
>
> ## Similarity to Klein et al
> We assume by Klein et al you are referring to https://arxiv.org/pdf/2302.01170.pdf.
>
> We are aware of this work and indeed it was an inspiration for our work, but we do not think it solves the problems that we set out to tackle in our research. In particular:
>
>  - Klein et. al. learn their models from samples only, while we pursue energy-based learning. Our contributions are all targeted towards making energy-based learning tractable.
>
>  - Klein et. al. use a non-rotation-equivariant flow architecture. They use data augmentation to learn a rotation invariant density. Our key contribution is the introduction of a novel SE(3) equivariant flow architecture.
>
>  - Klein et. al. learn conditional metropolis proposals to speed up existing MD schemes. We target the full, unconditional, Boltzmann distribution directly.
>
> In terms of similarities, we see the largest one as both papers using coupling flows with augmented variables to obtain permutation equivariance. Could the reviewer clarify to which further similarities in both papers’ proposals they are referring to?
>
>
> ## Code release
> Our code is available in the supplementary material. We will also be releasing the code repository upon publication!
>
> ## KLD normalizer estimate
> For alanine dipeptide the KLD is an empirical KLD between the histograms/Ramachandran plots in Figure 3. Because this is simply the KLD between two categorical distributions, it already normalized. This would not be the KLD between the flow and (unnormalized) target densities themselves, which which we do not report.
>
> ## Assessment of mode collapse from training by reverse KL.
> We do not perform training via reverse KL. Specifically for training-by-energy we use the Flow Annealed Importance Sampling (FAB) training technique, which encourages mass covering rather than mode seeking. We can see that the models do not suffer from mode collapse by their test set likelihood (which becomes very low if mode collapse occurs). The models trained with FAB have a similar (or better) test set likelihood than the models trained by maximum likelihood - hence we can see that these models are fitting the target well. Forward ESS would be a good additional metric, however this requires the normalized target density where we only have the unnormalized density.
>
> ## Numerical instability of Cartesian Projection
> If vectors output from EGNN become collinear then the orthonormalization becomes numerically unstable. We use an additional loss that encourages non-collinear output vectors to mitigate this. It is described in Appendix B3.
>
> ## Dedicated limitations section
> We agree - hence we have added a limitations subsection to the discussion. This is given in our [general response to reviewers](https://openreview.net/forum?id=KKxO6wwx8p&noteId=Narle0scqP).

---

> > ### Comment · Reviewer_Xvtr · 2023-08-11
> >
> > Thank you for the detailed rebuttal. Overall I find your reply convincing and I am happy to hear that you will make the code publicly available.
> >
> > However, I have two follow-up questions/remarks:
> >
> > ### KLD estimator
> >
> > Is there any reason why you do not report ESS with respect to the target density? This seems like a relevant quantity to report.
> >
> > ### Mode Collapse
> > The statement that forward ESS requires a normalized density is incorrect. One can simply estimate the partition function by MC estimate of the expectation value of the unnormalized importance weights
> >
> > $$ Z^{-1} = E_p [ q(x) / exp(- \beta E(x) ] $$
> >
> > where $q$ is the model density and $p(x) = 1/Z exp(-beta E(x) $ is the target density. See Eq (22) in 2107.00734 for a discussion. This in turn can be used to define an estimator for the forward ESS (in complete analogy to the reverse ESS estimator).

---

> > > ### Author Response · Authors · 2023-08-14
> > >
> > > Thank you for the further feedback.
> > >
> > > You are indeed correct that with the formula you provided we are able to estimate $Z^{-1}$, which allows us to estimate the (joint) forward ESS with the test set data. We were not aware of this previously - thank you for pointing it out!
> > >
> > > We have evaluated the forward ESS of models trained with FAB, and the results are in fairly close agreement with the reverse ESS estimates.
> > > Since paper submission we have retrained all the models (notably we train LJ13 for longer with FAB which significantly improves results). In the table below we provide these results, with the forward ESS included:
> > >
> > > | Problem        | DW4         | DW4        | DW4 | LJ13   | LJ13            | LJ13 |
> > > |-------------------|-------|-------|------|------|-------|-------|
> > > |           | Rev ESS (\%)        | Fwd ESS (\%)           | NLL            | Rev ESS (\%)        | Fwd ESS (\%)           | NLL            |
> > > | **Non-E-ACF**           | 35.94 |  5.45 | 7.38  | 5.38| 4.14 | 33.22   |
> > > | **Vector-proj E-ACF**  | **84.29** | **83.39** | **7.11** | 59.60 | 65.20 | 30.33 |
> > > |  **Cartesian-proj E-ACF**| 82.44  | 80.08  | 7.13 | 60.68 | 65.54 | 30.34  |
> > > | **Spherical-proj E-ACF**   | 80.44  | 81.46  | 7.14 | **62.09** | **66.13** | **30.21**  |
> > >
> > >
> > > With regards to the ESS for alanine dipeptide w.r.t the (joint) target density - we agree that this would be a useful addition, and will add it to the camera ready version of the paper. Our expectation is that the ESS of all models on alanine dipeptide will be relatively poor - as training by maximum likelihood does not typically lead to good ESS - this is shown in https://arxiv.org/abs/2208.01893 for a flow on internal coordinates, and can be seen for the DW4 and LJ13 problems in Appendix C2 Table 5.

---

> > > > ### Comment · Reviewer_Xvtr · 2023-08-17
> > > > **Thank you for the reply.**
> > > >
> > > > Thank you very much for the reply. It is good go see that the there seems to be no evidence for mode-dropping. Interestingly, the Non-E-ACF flow seem to show some degree of mode dropping.
> > > >
> > > > I agree that the ESS for alanine dipeptide will be low. Nevertheless, it would be interesting to quantify this. Adding this to the Supplement would be valuable.

---

### Official Review · Reviewer_2cbB · 2023-07-02

**Soundness:** 3 good
**Presentation:** 4 excellent
**Contribution:** 2 fair
**Rating:** 7
**Confidence:** 3

**Summary:**

The paper proposes a method of producing a special Euclidean group - SE(3) - equivariant method of training coupling flows by augmenting the input space then conducting standard normalizing flow transformations in a rotationally invariant space by means of an invertible transformation. This is applicable particularly to Boltzmann distribution problems.

Update: I have read the rebuttal, and the additional details seem convincing. I have raised my score accordingly.

**Strengths:**

The proposed SE(3) Equivariant Flows offer a specialized framework for dealing with atomical problems which utilize SE(3) structure. For the task of approximating the Boltzmann distribution, the model works quite well compared to some of the baselines established, and features higher speedup. In addition, it admits a FAB-version of training the model without maximum likelihood.

**Weaknesses:**

Seeing the disparity between the NON E-ACF and 	the other models on the QM9 dataset, I would have liked to see some comparisons against the E-CNF DIFF baseline on the Alanine dipeptide baseline as well.

In addition, the baseline speedup comparison is only against the diffusion model for which they claim a 500x speedup. Though this speedup might be due to the advantages proposed by the framework, a large portion of it might be due to the denoising process intrinsically taking up a lot of time. I would like to see the authors either talk about speedups compared to other standard CNFs, or along the lines of a performance/speed tradeoff that can be made with respect to the results shown in the tables.

**Questions:**

In the paper, the primary baselines and comparisons are against CNFs, primarily due to the fact that existing literature on SE(3) equivariant flows have mainly through the CNF regime (though the authors do also mention equivariant residual flows as well). How does enforcing equivariance vs not enforcing affect subsequent performance on the benchmarks in the case of coupling flows?

**Limitations:**

Yes.

---

> ### Author Rebuttal · Authors · 2023-08-08
>
> Thanks for your feedback! Below we have responded to your comments.
>
> ## Diffusion on Alanine Dipeptide
> We agree that this would be a good additional baseline. Hence, we will add it to the camera ready version of the paper.
>
> ## Speedup of CNF
> The reviewer is correct that the large relative speedup is due to "to the (diffusion) denoising process intrinsically taking up a lot of time". Diffusion and discrete normalising flows tradeoff flexibilty for speed; diffusion is more flexible but discrete flows are faster. We will clarify this in the main text as it is one of the main motivations for our work.
>
> We evaluate the model trained with diffusion as a CNF (i.e. ODE not SDE) and use arguably the most standard CNF ode solver (dopri5) for evaluating the speed of the CNF sample generation. We agree that this could in principle be made faster (e.g. via Optimal transport flow matching or via trading performance for speed). However, we consider this to be outside of the scope of the paper.
>
> ## Effect of enforcing vs not enforcing equivariance
> We study the effect of enforcing vs not enforcing equivariance with our non-$SO3$ equivariant coupling flow baseline (NON-E-ACF in tables 1, 2 and 3). We fine that it consistently performs significantly worse than the equivariant models. Additionally, we have now added experiments comparing the equivariant models and non-equivariant models for different dataset sizes. We find that the equivariant models are substantially more data-efficient. These new results are summarised in our [general response to reviewers](https://openreview.net/forum?id=KKxO6wwx8p&noteId=Narle0scqP).

---

### Official Review · Reviewer_JYCH · 2023-07-04

**Soundness:** 3 good
**Presentation:** 3 good
**Contribution:** 2 fair
**Rating:** 6
**Confidence:** 4

**Summary:**

This paper proposes a novel method for developing an SE(3) equivariant coupling flow that works on the Cartesian coordinates of atoms, crucially enabling quick sampling and density evaluation. The standard coupling architecture is unable to operate on the Cartesian coordinates with the Special Euclidean group SE(3) and permutation invariances of physical systems, hence the researchers introduce an advanced coupling flow that preserves these by using coordinate splits along added augmented dimensions. The method includes mapping atoms’ positions into learned SE(3) invariant bases at each layer, applying standard flow transformations such as monotonic rational-quadratic splines, and then reverting to the original basis. This proposed model demonstrated competitive performance when trained on the DW4, LJ13, and QM9-positional datasets, offering much faster sampling speeds. The researchers also became the first to learn the full Boltzmann distribution of alanine dipeptide by only modeling the Cartesian positions of its atoms.

Furthermore, the study introduces a new approach to making coupling layers SE(3) equivariant by augmenting the input space with auxiliary variables that can be acted upon by SE(3). This allows the atom positions to be updated based on these auxiliary variables by first projecting the atoms into an SE(3)-invariant space, applying a standard normalizing flow transform and then projecting its output back onto the equivariant space. The researchers showcased that when trained by maximum likelihood, their flow matches the performance of both existing SE(3) CNFs and coupling flows operating on internal coordinates on molecular generation tasks. This flow is over 100 times faster for sampling than SE(3) CNFs. The study also demonstrates the flow's ability to be used in the energy-based training setting on the DW4 and LJ13 problems, showcasing its unique advantages over existing methods.

**Strengths:**

This paper has a few strengths that I think are worth mentioning:
1. The overall idea of using an augmented normalizing flow and baking in symmetries in the product space is a novel direction and sensible idea.

2. In general the presentation of the paper is of high quality (except a few notational quirks). The writing is easy to follow and is supplanted by a great architecture diagram in Fig 1.

3. The experimental results show a lot of promise and validate the design of the flow.

**Weaknesses:**




**Technical Details**

I had a difficult time following the proof of equivariance it uses slightly non-standard notation for defining the group action and how it works. Specifically, if $\theta$ is invariant and the standard flow is invariant I'm not sure how this could lead to an equivariant flow. That is we often require an equivariant pushforward to map an invariant base density to an invariant target density. There are two ways you could convince me: 1.) Randomly sample rotations from $g \in SO(3)$ and numerically prove equivariance of the overall flow is satisfied. 2.) Or you could provide a lot more detail and use more standard notation. For example, I think you mean $\tau_{\theta}$ is an invariant density but is actually an equivariant pushforward map. Otherwise, I'm not sure how this goes through.

The idea of using augmented variables is an interesting algorithmic choice, but using a standard coupling flow for $\tau_{\theta}$ is only 1 option. In fact, I believe you could use any flow architecture for $\tau_{\theta}$ as such it would be great to see if other mainstay flow architectures could be employed here (e.g. ResFlow, MintNet, BNAF).

**Experiments**
The biggest question I have from the experiments section is from Tab 1. It seems for DW4 training with samples is not competitive with E-CNF, but with training with energies, we find that the proposed approach clearly beats E-CNF. Can the authors provide more intuition and justification here?

One of the main benefits of using symmetries as an inductive bias is that it leads to increased sample efficiency. This is not explored in the current manuscript. One could consider an additional experiment with samples where an equivariant model and a non-equivariant model are trained with various different dataset sizes. One would expect the equivariant model to be more performant in the low data regime. Can the authors try this experiment?

**Minor**
- In the text you use CoM-Shift but in equations, this is called ShiftCoM. Better consistency in naming conventions should improve readability.

**Questions:**

1. Can you provide more details on the graph neural network $h$ in the main paper. It is not clear to me how this network outputs invariant parameters $\theta$. Is this the sum of all the node embeddings?

2. It is not clear to me how exactly the GNN $h$ outputs equivariant features. Are you using an EGNN and taking the velocity vectors? More detail on this in the main paper would be appreciated.

3. In Fig 1. The $\oplus$ operation is actually a concatenation rather than an addition right? This could be made more clear.

4. In line 210 it is stated "We first subtract the observations’ CoM from both the observed and augmented variable", but previously in line  134 it is stated, "Importantly, we do not restrict a to be zero-CoM." This appears to be a contradiction?

---

> ### Author Rebuttal · Authors · 2023-08-08
>
> Thanks for your review! Below we have responded to your feedback and questions.
>
> ## Proof of equivarance
> Indeed in our code we do test equivariance of our flow by testing that  $g \cdot F(x, a) = F(g \cdot x, g \cdot a)$ (and that the log det is invariant).
>
> We do not think that our notation for group action is non-standard. For instance https://arxiv.org/abs/2302.02277 uses very similar notation. In particular, we use $g$ to refer to a member of a group and $g \cdot x$ to refer to it being applied to a particular object $x$. We do not denote the representations of $g$ which act on different spaces separately, as these can always be distinguished by looking at the object upon which $g$ acts. Could the reviewer please clarify which specific notation is non-standard?
>
> We have updated the methods section to make it more clear and precise, specifically with respect to the definition of $\gamma$. This update is described in our [general response to reviewers](https://openreview.net/forum?id=KKxO6wwx8p&noteId=Narle0scqP).
>
> Let us try to further clarify how our coupling layer is constructed.
> $\tau$ is any bijective transformation, it is neither required to be invariant nor equivariant.
> We apply $\tau$ to features that live in an “invariant space”, that is, the outputs of $\gamma_r$.
>
> The map $\gamma: \psi \times X \rightarrow Y$ is invariant, meaning that $\gamma_{g \cdot r} \cdot g \cdot  x = \gamma_{r} \cdot  x^{\ell}$. Any rotations applied to $x$ (and $r$) will be undone by $\gamma$.
>
> After the bijection $\tau$ is applied, we apply the inverse of $\gamma_{r}$. This re-introduces any rotations present in $x$ (and $r$) that were undone by  $\gamma_{r}$ in the first place, resulting in an equivariant map.
>
> We do note that the wording is a bit tricky - specifically the density of the flow and base distribution is **invariant** but the flow pushforward is an **equivariant** map. We have tried to make this distinction clear in the text.
>
>
> ## Using other flows besides coupling flows
> Coupling flows are fast for both sampling and density evaluation (i.e. fast for both forwards transform and inversion). This is critical for our application of interest, where a flow that has fast density evaluation and sampling can be used for energy based training.
> We mention residual flows in the discussion noting that they are limited for our purposes by slow inversion. Similarly, the MintNet and Autoregressive Flows either have a slow forwards or inverse transform, hence are not well suited for application of interest.
>
>
> ## Why better than CNF when trained with energy but not when with samples?
> We are not certain that we are understanding the Reviewer's question here, we interpret is as asking why the E-CNF trained by maximum likelihood is worse than the E-ACF trained by energy, even though it is better than the E-ACF trained by samples. Could the reviewer please clarify the question if this is not the case.
>
> Generally, training by energy which uses ground truth density is better than training on a small (possibly biased) dataset from MD simulation, as training by energy results in the flow fitting the ground truth target function very well, where training by maximum likelihood is limited by the size and quality of the training dataset.
> Hence, the E-ACF trained by energy on DW4 obtains far better performance than when trained by maximum likelihood. The comparison between the E-CNF trained with samples, and the E-ACF trained by energy is not apples-to-apples, as both the training method and model type are being changes. However, overall this difference is mainly explained by the aforementioned benifits of training-by-energy, rather than differences in the models expresiveness.
>
>
> ## Testing sample efficiency with differerent dataset sizes
> We agree this is an interesting question. Hence, we tested this for DW4, LJ13 and Alanine Dipeptide. We summarized these results in the [general response to reviewers](https://openreview.net/forum?id=KKxO6wwx8p&noteId=Narle0scqP). Generally, we found that the equivariant flows are significantly more data efficient than the non-equivariant flows.
>
>
> ## COMShift vs ShiftCOM inconsistency
> Thanks for highlighting this! We have fixed it in the text.
>
>
> ## Details of EGNN
> The EGNN from sattoras et al, which has become a standard architecture in the field, outputs equivariant vectors and invariant scalars per node, see equations 3-7 in https://arxiv.org/abs/2102.09844. We use this design essentially verbatim (with a few minor tweaks that we have added to Appendix C.2.), where the input is the cartesian coordinates and node embeddings of the atoms. The invariant parameters are per node (not global) hence there is no pooling over nodes.
>
> ## Figure 1 concatenation instead of addition?
> This should be a subtraction (see Equation 5). We have fixed this in the image.
>
> ## Line 210 seems to imply restricting both $x$ and $a$ to zero-CoM?
> Given observations $x$ and augmented variables $a$, we first perform $(x - \bar{x}, a - \bar{x})$.
> This restricts $x$ to the Zero-Com subspace but crucially does not restrict $a$ to be ZeroCom. We agree that this was ambiguous in the text and have edited it to make this more clear.

---

> > ### Comment · Reviewer_JYCH · 2023-08-19
> > **Re: Rebuttal**
> >
> > I thank the authors for their responses.
> >
> > The notation is clearer in my mind now. I still contest that there are more clear ways of defining the group action which is in more line with the rest of the literature but that's not important.
> >
> > **Biggest Question**
> >
> > My biggest question, which I think the authors misunderstood is why in Table 1. do we see for DW4 E-CNF is way better (it's bolded) than your method? This is also true for the global response in the low data regime. I do not buy the biased MD simulations aspect here because it's orthogonal since you have a dataset---and are not training using energies. This falls perfectly into the current generative modeling setup and Maximum Likelihood framework. Thus I also disagree with the authors on the need for MintNet and ResidualFlows. It is true that coupling flows have fast inference but everyone in the literature knows that they have subpar performance---despite being universal density approximators. At present, I do not know how performant the presented flows are compared to E-CNF.
> > Furthermore, to demonstrate scalability the experiments would need to be conducted on a much larger dataset which is not done in the current work, so while this claim might be true it is not demonstrated.

---

> > > ### Author Response · Authors · 2023-08-20
> > >
> > > We note that the DW4 and LJ13 datasets are indeed generated by MCMC and are biased (see http://proceedings.mlr.press/v119/kohler20a/kohler20a.pdf where this dataset is introduced).
> > > In the very recent work https://arxiv.org/pdf/2306.15030.pdf choose to regenerative the training and test sets for DW4 and LJ13 as they noted that these were biased - although their datasets are not yet available publicly.
> > > For DW4 the difference in the empirical distribution of train samples and test samples is clearly visible in a histogram of the interatomic distances between the training and test sets.
> > > We will add this plot, and further clarification explaining this to the camera-ready version of the paper.
> > > This bias prevents accurate test log likelihood comparision and therefore these should be taken with a grain of salt.
> > > Additionally we note that when using the large dataset for DW4 containing $10^5$ samples that the E-ACF achieves better results than the E-CNF. Hence we can see that the higher NLL for DW4 for the E-ACF is not due to expresiveness limitations.
> > > The DW4 problem is very simple and we chose to include it as it is used in the foundational papers on equivariant flows.
> > > Instead of doing further investigation into the difference in NLL on DW4 (e.g. by creating our own training and test sets that were not biased), we felt that it would be better to focus on NLL comparisons on more complex problems, such as QM9-positional, where we see that our flow does well.
> > >
> > > MintNet and ResidualFlows are not more expressive than the CNF baseline used in this paper. Hence, we do not feel that their inclusion is critical - although of course more baselines are always better.
> > > With regards to scalability, we note that our E-ACF is not more scalable than CNF/ResidualFlows/MintNet for training with samples, and is only more efficient for training that includes both sampling and density evaluation in the training inner loop. Specifically this is needed for training Boltzmann generators by energy, which we demonstrate is possible with the E-ACF in Section 4.3, where training a CNF is intractable.
> > >
> > > We hope that this answers your concerns - please let us know if you have any remaining questions/concerns.

---

### Official Review · Reviewer_mxfv · 2023-07-07

**Soundness:** 3 good
**Presentation:** 2 fair
**Contribution:** 3 good
**Rating:** 7
**Confidence:** 2

**Summary:**

This paper tackles the problem of sampling molecule configurations using deep generative models. The learned distribution should be permutation invariant (in terms of permutations of the particles in $\mathbb{R}^3$) and SE(3) invariant (i.e., invariant to rotations and translations applied to the particles' cartestian coordinates). At the same time, the deep generative models should be fast to sample from and allow quick density evaluation.

Continuous normalizing flows (and relatedly, diffusion models) can be built with these symmetry constraints, but are slow to sample from. Coupling flows are fast at sampling/density evaluation but existing approaches rely on internal coordinates rather than explicit cartesian coordinates. Generally, the goal of this paper is to endow coupling flows with the SE(3) and permutation symmetries of existing CNFs, while preserving the speed benefits of coupling flows.

Coupling transforms pose a problem for SE(3)xS_n symmetry because splitting the coordinates will break either rotation or permutation symmetry, depending on which dimension one splits on. The paper gets around this problem by introducing augmented dimensions, which are instantiated as additional observations $a$. The original observations are transformed conditioned on the augmented variables and vice versa, with equivariance enforced by having a particular structure to the transformation of x given a (Eq 4). In experiments, the paper shows that the proposed method performs similarly to CNF based methods while preserving the computation benefits of coupling flows.

**Strengths:**

This paper presents a solid and principled design for making coupling layers (in a normalizing flow) SE(3) and permutation equivariant, a property of practical interest in situations such as modeling molecular configurations. The experiments show that their approach preserves the computation benefits of coupling flows and the performance benefits of symmetry priors.

**Weaknesses:**

Although the preliminaries are set up well, the presentation of the method starting from 3.1 is somewhat confusing and could benefit from more intuitive description (eg, of Prop 3.2) and concrete examples.

**Questions:**

N/A

**Limitations:**

The paper is not particularly clear on limitations of the proposed approach.

---

> ### Author Rebuttal · Authors · 2023-08-08
>
> Thank you for your review! We have addressed the issues you highlighted below.
>
> ## Clarity of methods section
> We have updated the description of our method to make it more clear. Additionally [Reviewer goAi](https://openreview.net/forum?id=KKxO6wwx8p&noteId=fraAiN4VoH) found a typo in one of the equations in the text for this section which we have now fixed. The updated description of the method can be seen in the [general response to reviewers](https://openreview.net/forum?id=KKxO6wwx8p&noteId=Narle0scqP).  We now briefly note in the text an example $\gamma\_r$ defined by a projection into a frame composed of a rotation matrix and origin. Due to space constraints we are currently unable to fit a more detailed description illustrating how each of the components of the flow work.  However, Figure 1 provides a (very high level) overview of the flow steps, while Appendix B.3 goes through the various projection types in detail.
>
> We hope to add further a more detailed example of $\gamma$ to the main text if we can make more space. We provide this example below - we hope it makes the projection-into-an-invariant-space component of the flow more clear.
>
> >  The Cartesian projection into the invariant space is given by $x_{invariant} = R^{-1} (x_{equivariant} - o)$ where the projection is composed by an orthonormal rotation matrix R and the origin o, where both R and o are equivariant.  The inverse of this projection is then simply $x_{equivariant} = R x_{invariant} + o$.
>
>
> ## Unclear limitations
> We agree that the limitations were not made sufficiently clear, hence we have added an explicit limitations subsection to our discussion. This new section is provided in [general response to reviewers](https://openreview.net/forum?id=KKxO6wwx8p&noteId=Narle0scqP).

---

### Official Review · Reviewer_goAi · 2023-07-14

**Soundness:** 3 good
**Presentation:** 1 poor
**Contribution:** 2 fair
**Rating:** 5
**Confidence:** 3

**Summary:**

This paper constructs equivariant normalizing flows that are discrete to sidestep speed issues with equivariant continuous normalizing flows. The method does this by augmenting the input with some an additional variable to avoid representation issues.

**Strengths:**

* The method is a reasonable extension of standard equivariant normalizing flows literature that seeks to address the fundamental problem of balancing expressivity and speed.
* The method produces good results (ie likelihoods and other metrics are trained to a very good value while not using much compute).

**Weaknesses:**

* I still don't understand the usage of $\gamma_r$. Line 160-161 says that $\gamma_r$ is a projection to an invariant space, but the definitions in Line 162 don't seem to accurately describe this (which should be $\gamma_r \cdot gx = \gamma_r \cdot x$ if I'm correct. Also, what is the definition of $\gamma_r^{-1}$, since I assume that necessarily $\gamma_r$ must got a lower-dimensional subspace. It would also help tremendously if the "Choice of projection $\gamma$" section was written to be more mathematically precise.
* This "invariant feature projection" that $\gamma_r$ is known to be a rather weak form of constructing invariant functions. Some expressivity proof would be helpful to allay these types of concerns.
* If $\gamma_r$ projects to something like S^2 angles, which I think might be happening for sphere-proj, then wouldn't one need a manifold-based flow model to accurately build a flow on this? See e.g. [1, 2].
* To evaluate densities, the paper computes using only 20 samples for a. This seems like a low amount.
* Is there a reason why results aren't reported for DW4 or LJ13 for diffusion in table 1? I assume this is trained from samples, which should allow for diffusion.
* In the appendix, diffusion seems to be trained for a very short amount of time (less than even the for the proposed fast ACF methods. From the literature, diffusion tends to take a while to train but this tends to improve log-likelihood.
* For table 2, why can't the negative log-likelihood be compared? Also I think its KL divergence is lower than the proposed method (at least from the table).

[1] https://arxiv.org/abs/2002.02428
[2] https://arxiv.org/abs/2006.10254

**Questions:**

Nothing outside of the listed weaknesses.

**Limitations:**

Yes

---

> ### Author Rebuttal · Authors · 2023-08-08
>
> Thank you for your feedback! We have addressed each of the weaknesses you highlighted below. We hope that this helps clarify the paper.
> ## Usage of $\gamma$:
> You are correct; there is a typo in the text - thanks for pointing this out. The property of interest is $\gamma_{g \cdot r} = \gamma_{ r} \cdot g^{-1}$. Furthermore, we have updated text to make the definition of $\gamma$ more clear. We have highlighted these changes in our [general response to reviewers](https://openreview.net/forum?id=KKxO6wwx8p&noteId=Narle0scqP).
>
> In the case of the Cartesian and spherical projections the invariant space is the same dimension as the equivariant space. The vector projection invariant space is lower dimensional, but this can simply be seen as a case of the spherical projection where the two angles are held constant. The projections are standard - hence we relegate their details to the Appendix B3. For example, in the case of the Cartesian projection the projection into the invariant space is given by $x_{invariant} = R^{-1} (x_{equivariant} - o)$ where the projection is composed by an orthonormal rotation matrix R and the origin o, where both R and o are equivariant.  The inverse of this projection is then simply $x_{equivariant} = R x_{invariant} + o$. We hope this example makes $\gamma$ more clear.
>
> ## Expressivity limitations of invariant projection
> Invariant projections have been quite successful for expressive transformations of equivariant spaces. Specifically the Invariant Point Attention (IPA) used in AlphaFold 2 uses a similar idea of projection into a local invariant frame associated with each protein residue. The IPA network was also used in frame diffusion (https://arxiv.org/abs/2302.02277). This similarity has now been noted in the Discussion.
>
> As the invariant features are of the same dimensionality as the equivariant features, information is not lost in the projection. One similar case which may be causing some confusion is GNNs that use lower dimensional representations (such as distances between points instead of vectors), which is very harmful for expressiveness.
> Empirically, we are able to capture complex functions, like QM9 and alanine density, which are challenging even for flexible diffusion models.
>
> ## Manifold flow for angles
> Yes, for the spherical projection we have one angle bounded between $[0, \pi]$ and another between $[-\pi, \pi]$ and we indeed use the flows from [1] that you listed (which we also cite). This is described in Appendix B.3.
> ## Number of samples of $a$ for marginal density estimation
> Empirically we find that the marginal log likelihood plateaus around this value. For example, on DW4 we tested increasing from 20 to 50 samples of $a$ per $x$ changes the marginal log likelihood is 0.01 (tested on a single trained model) and for LJ13 it made a 0.03 difference. Setting the number of samples to 20 allows us to obtain good estimates of the marginal log likelihood, while preventing the evaluation costs from becoming unnecessarily large.
>
> As we discuss in line 244, “evaluating densities”, the variance of our density estimator vanishes when our flow is able to match the augmented target distribution exactly. This distribution is a simple Gaussian, which is often modelled well by out flow. This result explains why we can get away with a relatively small number of samples.
> ## Why no diffusion for DW4 and LJ13?
> On these datasets we report the CNF results of (Satorras et. al.). This model is equivalent to modern diffusion models, although the authors train directly by likelihood instead of score matching.
> Although we did train diffusion models on these datasets our results were poor for these problems (i.e. worse than the CNF results of (Satorras et. al.)).
> The models for these problems therefore require further fine-tuning which we were not able to complete before the deadline.
> Specifically, the noise schedule requires further tuning as we found it to dramatically effect performance.
> We will add results with the CNF-Diffusion baseline for the camera-ready version of the paper. We expect it to get similar results to the E-CNF for these problems.
> On QM9 positional, we were able to substantially improve upon the results of Satorras et. al. with our diffusion models. This is likely because  Satorras et. al. were unable to train to convergence. Thus we report the results for both CNFs, with CNF-Diff representing the strong baseline.
> ## Diffusion trained for short
> We found that the biggest difference for training diffusion was optimizing the networks and noise schedule, while further increases in the training time did not make a large difference.
> However, for the camera ready version of the paper we look into the effect of training the diffusion model for a much larger training time.
>
> The speed of our flow is advantageuous compared to CNFs for training that requires density evaluation or sampling. This applies to training by maximum likelihood and energy-based training. Hence, in the maximum likelihood regime our flow trains for much shorter than that of Satorras et. al.
> However, the score-matching objective used in diffusion does not require density evaluation and is highly parellelizable.
> Thus, in the training-by-samples regime we do not expect our flow to be faster for training than diffusion models.
>
> ## Comparing log likelihood on Aldp, KLD of histograms
> The spaces on which densities are defined are different on internal coordinates and on 3D coordinates making their NLL not comparable.
> You are correct that the flow on internal coordinates does better on the KL metric than our method. This is consistent with Figure 3, the histogram from which this empirical KLD is calculated, where we see that the internal coordinates match the data the closest. The internal coordinates represent a very favourable but non-transferable representation of a molecule’s conformations.

---

> > ### Comment · Reviewer_goAi · 2023-08-15
> > **Further questions**
> >
> > Thank you for the clarifications.
> >
> > ## On Likelihoods
> >
> > Why does the choice of space make the likelihoods incomparable? Is it fundamentally mathematical, or more for a comparison purposes. It seems like they should be comparable (since the internal coordinate model is just the 3d coordinate system modulo symmetries).

---

> > > ### Author Response · Authors · 2023-08-15
> > >
> > > The internal coordinate model has the following key differences which prevents us from directly comparing the LL
> > >  - It is rotation invariant (global rotation does not change the internal coordinates). This means it lies on a lower dimensional manifold than the Cartesian coordinates / the mapping from Cartesian coordinates to internal coordinates is surjective.
> > >  - It is centred on the first atom unlike the Cartesian coordinates which are zero centre of mass.
> > >
> > > Theoretically, it should be possible to convert the density in internal coordinates to an equivalent in Cartesian space. However, this conversion is non-trivial and will have a factor that is dependant on the configuration (i.e. we cannot convert our internal coordinates model LL to a cartesian coordinates LL via a constant shift). We are looking into doing this aforementioned conversion, and expect to complete it for the camera ready version of the paper.

---

> > > > ### Author Response · Authors · 2023-08-17
> > > > **LL of internal coordinate flow is now comparable**
> > > >
> > > > We have now calculated the density of the internal coordinate flow in Cartesian space with Zero-CoM and degrees of freedom representing global rotation, such that it may be compared with the other flows in our paper. With this adjustment the NLL of the flow on internal coordinate flow is -190.16, where the best E-ACF flow's NLL is −188.61. Thus, the internal coordinate flow performs slightly better, which is consistent with the Ramachandran plot (Figure 3), and the empirical KLD where the internal coordinate flow also performs slightly better.
> > > >
> > > > Note that although the model on internal coordinates performs slightly better, it wouldn't be able to scale well to higher dimensions since internal coordinates capture long range interactions poorly, while Cartesian coordinates capture them well. Furthermore, Cartesian coordinates form a more universal representation of molecules. They are the coordinates of choice when modelling the configurations of many molecules with one model.
> > > >
> > > > Below we provide a high level description of how we adjusted the density of the internal coordinate flow to make it comparable.
> > > >
> > > > ## Details on conversion of internal coordinate flow density.
> > > > In the original version of the paper, the internal coordinate flow included a transform that converted the internal coordinates into Cartesian coordinate space that had the following key differences to the space that the E-ACF is defined over.
> > > > - Their representation is invariant to global rotation, i.e. there are no degrees of freedom representing global rotation
> > > > - They are centered on the first atom, rather than having zero-CoM.
> > > >
> > > > We denote this representation of the flows coordinates as $z \in Z$.
> > > > We convert these into an equivalent density on the same space $X$ as what the E-ACF are defined over by (1) sampling a global rotation and applying this to $z$, and (2) converting the resultant coordinates to be zero-CoM.
> > > > Accounting for the probability of the rotation, and the Jacobian determinants of transforms (1) and (2) gives the following formula for the log density
> > > > $$\log q(x) = \log q(z)  - \log 8 \pi^2 -  2 \log(b_1) - \log(b_2) - \log(\sin(a_2)) + 3 \log n$$
> > > > where $b_1$ is the bond length between atom 0 and 1, $b_2$ is the bond length between atom 0 and 2,  $a_2$ is the angle between the first and second bonds, and $n$ is the number of atoms.

---

> > > > > ### Comment · Reviewer_goAi · 2023-08-17
> > > > > **Thanks**
> > > > >
> > > > > Thanks for all of the additional experiments + information. Raising score.

---

### Author Rebuttal · Authors · 2023-08-08

# General response
Thank you for the reviews - these have been very helpful and we have incorporated the feedback into the paper. Below we highlight key changes to the paper: **(1)** Added new results studying the data-efficiency of the equivariant vs non-equivariant variants of the flow. **(2)** Improved the clarity of the methods section. **(3)** Added explicit limitations section.

## Testing sample efficiency with differerent dataset sizes
We have run additional experiments comparing the data efficiency of the equivariant vs non-equivariant models for DW4, LJ13 and Alanine dipeptide. These are summarized below.

**DW4 and LJ13:**
We vary the dataset sizes following Satorras et al. [2021a]. We see that the equivariant flow is more data efficient, with larger performance gaps between equivariant and non-equivariant flow variants in the lower data regime.


| Problem        | DW4         | DW4          | DW4          | DW4 | LJ13   | LJ13          | LJ13            | LJ13 |
|-------------------|-------|-------|------|------|-------|--------|-------|-------|
| **Training dataset size**          | $10^2$           | $10^3$            | $10^4$            | $10^5$  | $10$    | $10^2$           | $10^3$            | $10^4$ |
| **E-CNF**              | **8.31**  | **8.15**  | **7.69** | 7.48 | **33.12** | 30.99  | 30.56 | 30.41 |
| **Non-E-ACF**           | 10.24 | 10.07 | 8.23 | 7.39 | 40.21 | 34.83  | 33.32 | 31.94 |
| **Vector-proj E-ACF**  | 8.65  | 8.69  | 7.95 | 7.47 | 33.64 | **30.61** | **30.19** | **30.09** |
|  **Cartesian-proj E-ACF**| 8.79  | 8.82  | 7.93 | 7.52 | 36.66 | 31.72  | 30.89 | 30.48 |
| **Spherical-proj E-ACF**   | 8.66  | 8.61  | 7.97 | **7.38** | 34.79 | 31.26  | 30.33 | 30.16 |


**Alanine dipeptide:**
We varied the training dataset size from $10^2$ to $10^6$ by subsampling the original training dataset, which has $10^6$ datapoints. We report the NLL for each model, with the best, i.e. lowest, value marked in **bold**. For all dataset sizes the equivariant models clearly outperform the non-equivariant model. Notably, if we only use 10% (vector-proj) or even just 1% (Cartesian-proj and spherical-proj) of the original dataset, our equivariant models are still better then the non-equivariant one trained on the full dataset.


| Training dataset size    | $10^2$           | $10^3$            | $10^4$            | $10^5$            | $10^6$            |
| ------------------------ | ---------------- | ----------------- | ----------------- | ----------------- | ----------------- |
| **Non-E-ACF**            | $-28.8$          | $-76.8$           | $-120.4$          | $-164.3$          | $-177.8$          |
| **Vector-proj E-ACF**    | $-31.0$          | $-111.4$          | $-169.5$          | $-183.6$          | $-188.5$          |
| **Cartesian-proj E-ACF** | $\mathbf{-65.1}$ | $\mathbf{-155.7}$ | $\mathbf{-185.9}$ | $-188.1$          | $\mathbf{-188.6}$ |
| **Spherical-proj E-ACF** | $-53.1$          | $-119.3$          | $-183.6$          | $\mathbf{-188.3}$ | $\mathbf{-188.6}$ |



## Improving methods clarity.
We have edited the Methods section to make the definition of $\gamma$ more precise, additionally we fixed a typo in one of the equations on Line 160-161. Here is the language we now use:

> $SE3 \times S_n$ equivariant coupling:
We dissociate the equivariance constraint from the tractability of the Jacobian determinant by constructing a core coupling  transformation that composes (a) an invariant map $\gamma: \psi \times X \rightarrow Y$, where $Y$ is isomorphic to $X$, and $r \in \psi$ parametrises the map. We denote the parametrised map as $\gamma_r$, e.g., a frame composed of an origin and a rotation matrix (b) a standard normalizing flow transform $\tau: Y \rightarrow  Y$, \(c\) the inverse map $\gamma^{-1}$.
Denoting the inputs with superscript $\ell$ and with $\ell+1$ for outputs, our core transformation $F: (x^{\ell}; a^{\ell}) \mapsto (x^{\ell+1}, a^{\ell+1})$ is given as
$$ x^{\ell+1} = \gamma^{-1}_{r} \cdot \tau\_{\theta} ( \gamma\_{r} \cdot x^{\ell}  ), \quad \text{with} \quad (r, \theta) = h(a^{\ell}) \\ a^{\ell+1} = a^{\ell}. $$
Here, $h$ is a (graph) neural network that returns a set of equivariant reference vectors $r$ which parametrise the map $\gamma\_{r}$, and invariant parameters $\theta$. $Y$ is a rotation invariant space. This means that any rotations applied to the inputs will be cancelled by $\gamma\_{r}$, i.e. $\gamma\_{g \cdot r} = \gamma\_{ r} \cdot g^{-1}$ or equivalently $(\gamma\_{g \cdot r})^{-1} =  g \cdot \gamma^{-1}\_{r}$ for all $g \in SO3$.
We use the inverse projection $\gamma\_{r}^{-1}$ to map the invariant features back to equivariant features.
The function $\tau\_{\theta}$ is a standard flow transformation, that we apply to the invariant features.

## Limitations
We have added a limitations subsection to our discussion to make these more clear.
> **Limitations**: Although our flow is significantly faster than alternatives such as CNFs, the expensive EGNN forward pass required in each layer of the flow makes it more computationally expensive than flows on internal coordinates. Additionally, we found our flow to be less numerically stable than flows on internal coordinates, which we mitigate via adjustments to the loss, optimizer and EGNN network (see App. B.3, App. C.1, App. C.2). Our implementation uses the E(3) equivariant EGNN proposed by Satorras et al. [2021a]. However, recently there have been large efforts towards developing more expressive, efficient and stable EGNNs architectures [Fuchs et al., 2020, Batatia et al., 2022, Musaelian et al., 2023, Liao and Smidt, 2023]. Incorporating these into our flow may improve performance, efficiency and stability. This would be especially useful for energy-based training, where the efficiency of the flow is a critical factor.

---

### Decision · Program_Chairs · 2023-09-21

**Decision:**

Accept (spotlight)

**Comment:**

Thank you for your submission and active engagement throughout the review period. The reviewers and I are all in agreement that the paper significantly advances the use of equivariant CNFs for modeling molecular configurations.